



# 1 Decadal trends (2013–2023) in PM10 sources and oxidative potential at a European urban supersite (Alpine Valley, Grenoble, France)

Vy Ngoc Thuy Dinh[1], Jean-Luc Jaffrezo[1], Pamela A. Dominutti[1], Rhabira Elazzouzi[1], Sophie
Darfeuil[1], Céline Voiron[1], Anouk Marsal[1], Stéphane Socquet[2], Gladys Mary[2], Julie Cozic[2],
Catherine Coulaud[1], Marc Durif[3,4], Olivier Favez[3,4], Gaëlle Uzu[1]
[1] Université Grenoble Alpes, CNRS, IRD, INP-G, INRAE, IGE (UMR 5001), 38000 Grenoble, France
[2] Atmo Auvergne-Rhône-Alpes (Atmo AuRA), 69500 Bron, France
[3] INERIS, Parc Technologique Alata, BP 2, 60550 Verneuil-en-Halatte, France
[4] Laboratoire central de surveillance de la qualité de l'air (LCSQA), 60550 Verneuil-en-Halatte, France
Correspondence to: Gaëlle Uzu *gaëlle.uzu@univ-grenoble-alpes.fr*

### 13 Abstract

The identification of particulate matter (PM) sources and the quantification of their contribution to the urban
environment is a necessary input for policymakers to reduce the air pollution impacts. The association between
the PM sources and the oxidative potential (OP) is also a key indicator for evaluating the ability of PM sources to
induce *in-vivo* oxidative stress and lead to adverse health effects, which becomes an emerging metric in the
Directive on ambient air quality (22024/2881/EU). Most studies in Europe have focused on PM and OP sources
in the short term, for only 1 or 2 years. However, the efficiency of reduction policies, trends, and epidemiological
impacts cannot be properly evaluated with such short-term studies due to a lack of statistical robustness. Here,
long-term $PM_{10}$ filter sampling at the Grenoble (France) urban background supersite and detailed chemical
analyses were used to investigate decadal trends of the main PM sources and related OP metrics. Positive matrix
factorization (PMF) analyses were conducted on the corresponding 11-year dataset (Jan 2013 to May 2023, n =
1570), enlightening the contributions of 10 PM sources: mineral dust, sulfate-rich, primary traffic, biomass
burning, primary biogenic, nitrate-rich, MSA-rich, aged sea salt, industrial and chloride-rich. The stability of the
chemical profile of these sources was validated by comparison with the profiles retrieved from shorter-term (3
years) successive PMF analyses. A Seasonal-Trend using LOESS decomposition was then applied to evaluate the
trends of these $PM_{10}$ sources, which revealed a substantial decrease in $PM_{10}$ (-0.73 µg m$^{-3}$ yr$^{-1}$) as well as that of
many of the $PM_{10}$ sources. Specifically, negative trends for primary traffic and biomass burning sources are
detected, with a reduction of 0.30 and 0.11 µg m$^{-3}$ yr$^{-1}$, respectively. The OP $PM_{10}$ source apportionment in 11
years confirmed the high redox activity of the anthropogenic sources, including biomass burning, industrial, and
primary traffic. Eventually, downward trends were also observed for $OP_{AA}$ and $OP_{DTT}$, mainly driven by the
reduction of residential heating and transport emissions, respectively.
Keywords: $PM_{10}$ source apportionment, OP $PM_{10}$ source apportionment, long-term trend, Positive matrix
factorization.



## 1. Introduction

Particulate matter (PM) is the main atmospheric pollutant that significantly impacts human health, climate, and the environment (Fuzzi et al., 2015; Grantz et al., 2003; Pope and Dockery, 2006), which is emitted directly or formed through complex processes in the atmosphere from natural and anthropogenic gaseous precursors. The identification of PM sources is important to investigate their composition, contribution, and evolution, which is one necessary input for policymakers to apply strategies in reducing their impact. There are various methodologies to identify these sources, where receptor models are widely used to perform source apportionment (SA) due to their flexibility and performance. Positive Matrix factorization (PMF) is one of the most popular among these receptor models, as it has been developed to allow SA analysis without any prior information other than the measurement and uncertainty input matrices (Hopke, 2016). Scores of studies using PMF have been applied in different typologies of sites over the last 15 years, with urban areas being the most common (Hopke et al., 2020; Viana et al., 2008).

The adverse health effects of PM can be assessed through different pathways, one of which uses the concept of oxidative stress within the lung (Pope and Dockery, 2006). PM has the ability to generate reactive oxygen species (ROS), which can cause an imbalance with antioxidants in the lungs, eventually causing oxidative stress. This capacity is evaluated as the oxidative potential (OP) of PM (Ayres et al., 2008; Li et al., 2008; Lodovici and Bigagli, 2011; Mudway et al., 2020; Nelin et al., 2012; Rao et al., 2018). The redox activity of PM is mainly dependent on their compositions; nevertheless, the correlation between individual components of PM and OP is probably not the best approach for understanding the impact of ambient PM because of their complex mixture preventing the quantification of all components of interest (Borlaza, 2021; Calas et al., 2018; Weber et al., 2018). Therefore, the relationship between OP and PM sources has been investigated as a more holistic approach (Bates et al., 2018; Dominutti et al., 2023; Weber et al., 2021). The implementation steps of such an approach can include, first, a PM source apportionment (SA) (usually using PMF), allowing the identification of PM sources and their contribution to PM. Then, the relationship between OP and PM sources is investigated by performing some regression techniques, potentially including linear and non-linear ones (Ngoc Thuy et al., 2024).

The OP of PM is becoming an emerging metric for the European regulation on air quality, included in the new European Air Quality Directive (Directive (EU) 2024/2881) as a recommended measurement at super sites in each member state in order to improve the knowledge about the variability of the OP and eventually allow the connections with epidemiological studies. Most previous studies have focused on PM and OP sources over a relatively short period, typically less than 1 or 2 years (Borlaza et al., 2022; Pietrodangelo et al., 2024; Weber et al., 2019). Such short-term studies assess the PM and OP sources as well as their contribution, providing information on the intrinsic OP of PM sources, allowing for the development of OP modeling (Daellenbach et al., 2020; Vida et al., 2024) and eventually designing some public policies (Borlaza, 2021). However, long-term series are needed both for evaluating the efficiency of such reduction policies in connection with the evolution of contributions from sources and also for implementing epidemiological studies (Borlaza-Lacoste et al., 2024).

The present study is based on a long-term measurement program conducted in the city of Grenoble (France), resulting from a sustained collaboration between the local network (Atmo AuRA), the French Reference Laboratory for Air Quality Monitoring (LCSQA), and the Institute of Environmental Geosciences (IGE) to investigate long-term evolution of $PM_{10}$ sources and OP in the $PM_{10}$ as well as their tendencies in the urban background of the city. Here, we assessed these source contributions from daily ambient $PM_{10}$ samples obtained



from 2013 to 2023 (n = 1570) using the EPA PMF model at this site selected as one of the French urban supersites
for the new EU 2024/2881 Air Quality Directive. The database was augmented using imputation techniques in
order to fill some of the gaps in the data, relative to metallic trace elements. Since PMF is rarely applied to such
a long-term database, several evaluations of the validity of solutions were also implemented. The PMF-derived
$PM_{10}$ sources were then used to perform OP SA from 2013 to 2022 (n=1570). The trend of $PM_{10}$ concentration,
of the $PM_{10}$ sources, and the OP measurements are eventually discussed in relation to several potential factors of
influence.
**2. Methodology**
**2.1. Sampling site**
$PM_{10}$ daily samples were collected at an urban background site (Grenoble - Les Frênes), in the southern area of
Grenoble, France (45°09′41″ N, 5°44′07″ E). This city is known as the French Alps' capital, sprawling over 18.13
$km^2$ with about 154,000 inhabitants in 2023, but nearly 500,000 within the larger urbanized area (about 15 km
radius). With an average altitude of about 200 masl, the city sits within a complex mountainous geomorphology
and is surrounded by three mountain massifs: Chartreuse, Vercors, and Belledonne (Figure 1). A pendular wind
regime between the three valleys of the basin regulates the ventilation of the atmosphere, with frequent thermal
inversion during cold periods, leading to the accumulation of pollutants. The air quality is monitored at several
sites in Grenoble by the regional agency (Atmo AuRA), including the urban background site of this study, which
is equipped with a large array of instruments. Particularly, the chemistry of $PM_{10}$ collected on filters has been
speciated at this site since 2008, within several programs, including the CARA program from the French Ministry
of Environment (Favez et al., 2021) and several research programs such as QAMECS (Borlaza et al., 2021), or
SOURCES (Weber et al., 2019). Many aspects of air quality in Grenoble were reported for this site, including the
characteristics of secondary anthropogenic PM fraction (Baduel et al., 2009, 2012; Favez et al., 2010; Tomaz et
al., 2016, 2017), of the biogenic PM components (Brighty et al., 2022; Samaké et al., 2019a, a), as well as the PM
OP (Borlaza, 2021; Dominutti et al., 2023; Weber et al., 2021). Several studies of one-year PM sources
apportionment were also performed in 2013 (Srivastava et al., 2018) and 2017-2018 (Borlaza et al., 2021). Despite
the difference in input data and periods of the studies, similar main sources of PM were quantified in both studies,
including residential heating, traffic, and secondary inorganic aerosol (SIA).



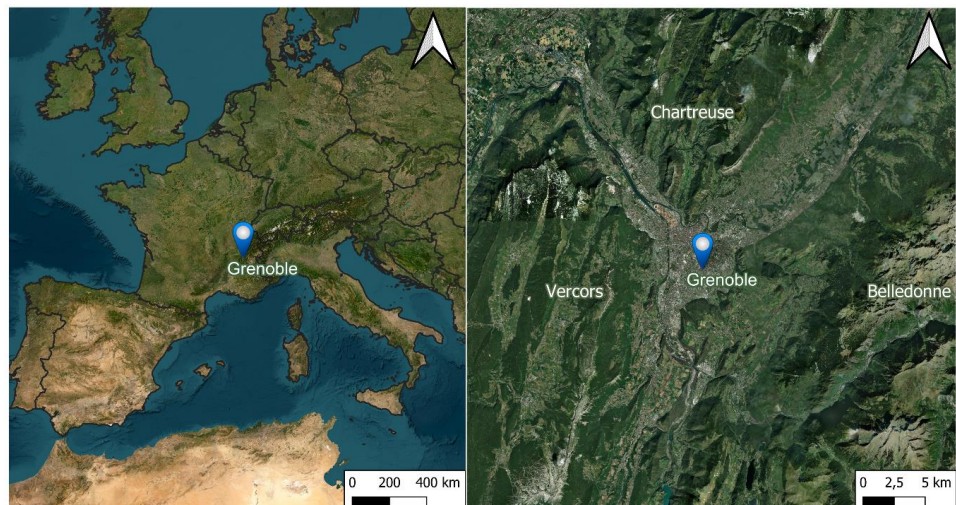

**Figure 1. The sampling site is located in the Southeast of France (left figure), surrounded by 3 mountains massifs (Vercors, Chartreuse, and Belledonne). Background map: ESRI satellites.**

**2.2. Sampling and chemical analyses**

**2.2.1. PM$_{10}$ and their inorganic and organic composition**

The daily PM$_{10}$ sampling was performed every third day from 02/01/2013 to 28/05/2023, on 150 mm-diameter quartz fibre filter (Tissu-quartz PALL QAT-UP 2500 diameter 150 mm) using high-volume samplers (Digitel DA80, 30 m$^3$ h$^{-1}$). A weekly PM$_{10}$ sampling was conducted during the same period using a low-volume sampler (Partisol, 1 m$^3$ h$^{-1}$) onto 47mm diameter quartz fibre filters (Tissuquartz PALL QAT-UP 2500 diameter 47 mm). The processes of preparation, handling, and storing filters, in order to guarantee optimum quality for chemical analyses were presented in Borlaza et al. (2021). Field blank filters were also collected (about 8-10% of the total samples) to estimate the detection limits and evaluate the filter contamination during the overall handling and analysis processes.

The daily PM$_{10}$ samples (n = 1570) and field blanks were analyzed for elemental carbon (EC) and organic carbon (OC), major ions (Cl$^-$, NO$_3^-$, SO$_4^{2-}$, Na$^+$, NH$_4^+$, K$^+$, Mg$^{2+}$, Ca$^{2+}$), methanesulfonic acid (MSA), anhydrous sugar and saccharides (levoglucosan, mannosan, arabitol, mannitol), and trace elements (As, Ba, Cd, Cr, Cu, Mn, Ni, Pb, Sb, V, Zn). However, the concentrations of the daily trace elements were analyzed only in 3 periods, including: (1) from January 2nd, 2013 to December 31st, 2013 (n = 122), (2) from February 28th, 2017 to March 13th, 2018 (n = 125), (3) from June 30th, 2020 to June 18$^{th}$, 2021 (n=115). The weekly samples and blanks were analyzed for trace metal concentrations for the whole sampling period (n = 842).

All analyses were previously described in detail (Borlaza et al., 2021). In brief, EC and OC analysis was performed using a Sunset Lab analyser with the EUSAAR2 thermo-optical protocol. The eight major ionic components and MSA were analyzed, after aqueous extraction of the filters using orbital shacking, by ionic chromatography using an ICS3000 dual-channel chromatograph (Thermo-Fisher) with a CS16 column for cation analysis and an AS11 HC column for anion analysis. The anhydrous-sugar and saccharides analyses were performed on the same water extract by an HPLC method using PAD (Pulsed Amperometric Detection) (model Dionex DX500 + ED40) with Metrosep columns (Carb 1-Guard+A Supp15-150+Carb1-150) in the period before the year 2017. From 2017 to



the present, the measurement with ICS 5000 with pulsed amperometric detection (HPLC-PAD) was performed
following the CEN method (European committee for standardization, 2024). The analysis is isocratic with 15%
eluent of sodium hydroxide (200 mM), sodium acetate (4 mM), and 85% water at 1 mL min$^{-1}$.
The daily and weekly metals were measured by Inductively coupled plasma mass spectroscopy (ICP-MS) (ELAN
6100 DRC II PerkinElmer or NEXION PerkinElmer). The measurement was performed on the mineralization of
a 38 mm diameter punch of each filter, using 5 mL of HNO3 (70 %) and 1.25 mL of $H_2O_2$ for 30 min at 180°C in
a microwave.

### 2.2.2. OP analysis

Two complementary OP assays, including the two probes ascorbic acid (AA) and dithiothreitol (DTT) were
performed on the same filters with $PM_{10}$ components analysis (from 02/01/2013 to 28/05/2023, n = 1570). Filter
samples are extracted using a simulated lung fluid during 1h15 at 37°C, pH 7.4, as described in Calas et al. (2017),
which creates a physiological environment for the extraction. The AA method quantifies the consumption of
ascorbic acid, an endogenous antioxidant in the lung, by PM and was described in Calas et al. (2017, 2018). The
reaction mixture (extract + AA) was transferred to UV-transparent 96-well plates (CELLSTAR, Greiner-Bio), and
the residual AA was measured at 265 nm with a TECAN Infinite M200 Pro spectrophotometer. The AA
consumption rate (nmol min$^{-1}$) reflects the capacity of $PM_{10}$ to catalyze electron transfer from AA to molecular
oxygen.
DTT assay relies on dithiothreitol, a chemical surrogate for cellular reducing agents, allowing for emulation of in
vivo interaction among $PM_{10}$ and biological reducing agents (for example, nicotinamide adenine dinucleotide
(NADH), nicotinamide adenine dinucleotide phosphate oxidase (NADPH)). After incubation of the PM
suspension within the lining fluid with DTT, the remaining DTT was titrated with 5,5′-dithiobis-(2-nitrobenzoic
acid) (DTNB) to form 5-mercapto-2-nitrobenzoic acid (TNB). Absorbance at 412 nm (TECAN Infinite M200 Pro)
in 96-well plates provided the concentration of unconsumed DTT, from which the DTT consumption rate
(nmol min$^{-1}$) was calculated. The batches were standardized using common external references to ensure
consistency between batches.
After analysis, the OP activities were blank subtracted and then normalized using the $PM_{10}$ mass concentration
and the sampling air volumes. The mass-normalized OP (OP$^m$, nmol min$^{-1}$ µg$^{-1}$) represents the intrinsic OP of 1µg
PM, while the volume-normalized OP (OP$^v$, nmol min$^{-1}$ m$^{-3}$) represents PM-derived OP per m$^3$ of air. Each sample
is analyzed in triplicate for AA and triplicate for DTT, respectively. Consequently, the OP values presented in the
study are the mean and the standard deviation of these replicates.

### 2.2.3. Vertical temperature and other ancillary measurements

Vertical temperature and humidity were measured every 30 minutes from November 2017 to May 2023 using
Tinytag TGP-4500 from Gemini Data Loggers. A Stevenson Type Screen protects each Tinytag loggers from
radiant heat (direct sunlight). Sensors are installed at a minimum of 3m from the ground. The measurements have
been performed at different elevations of the Bastille hill, located a few hundred meters from the city center
(5°43'37.0" E, 45°11'40.8" N), including z = 230, 309, 496, 916m altitudes.
Further, measurement of the $PM_{10}$ mass was conducted (hourly) using tapered element oscillating microbalances
equipped with filter dynamics measurement systems (TEOM-FDMS) at the same site as the filter collection. The





PM concentration used in this study is the 24-hour average concentration, which is associated with the days of
filter-based sample measurement (from 02/01/2013 to 28/05/2023).
**2.3. Multivariate imputation by chained equations (MICE)**
The daily concentration of metals was only accessed in some periods, with the number of samples being 362 of
the total of 1570 samples, which would severely limit the size of the inputs for the PMF processing. We used the
weekly concentration measured over the whole period to estimate the missing daily data using an imputation
method. The daily concentration of metals was imputed by using the MICE algorithm implemented with
multilinear regression (Azur et al., 2011). These values were modeled conditionally depending on the observed
values of the daily $PM_{10}$ and $PM_{10}$ components concentration (i.e., weekly concentration, $PM_{10}$, and $PM_{10}$
components concentration). These components are OC, EC, MSA, Levoglucosan, Mannosan, Polyols, $NO_3^-$, $SO_4^{2-}$
, $Na^+$, $NH_4^+$, $K^+$, $Mg^{2+}$, $Cl^-$, $Ca^{2+}$. The data preparation and imputation processes are implemented through 4 main
steps, as presented in S1 and Figure S1, Supplement. The validation of this imputation is shown in Table S1 and
Figure S2.
**2.4. Persistent inversions detection**
Thermal inversion occurs when the vertical temperature gradient between the ground-based and higher-altitude
stations is positive. However, this constraint is restrictive and limits thermal inversion detection, especially when
the calculation is on average daily temperature (Largeron and Staquet, 2016). Hence, the persistent inversion is
detected, as discussed in Largeron and Staquet (2016), for days with :
$$average \left(\frac{T_{916} - T_{230}}{\Delta z}\right)_{Daily} > Mean \left(\frac{T_{916} - T_{230}}{\Delta z}\right)_{Winter} \quad (1)$$

for more than 72 consecutive hours
with:
$T_{916} - T_{230}$ is the difference between temperature at ground-base station (z = 230m altitude) and at high-elevation
station (z = 916m);
$\Delta z$ is the difference between the height of high and low elevations;
$\frac{T_{916} - T_{230}}{\Delta z}$ : is the bulk temperature gradient between z = 230 and z = 916m;
$Mean \left(\frac{T_{916} - T_{230}}{\Delta z}\right)_{Winter}$ : is the mean bulk temperature gradient in wintertime (from November to March).
**2.5. Positive Matrix Factorisation (PMF)**
**2.5.1. PMF input**
EPA PMF 5.0 (Gary Norris et al., 2014) was used to identify and quantify the $PM_{10}$ sources based on the observed
concentrations and their related uncertainties. The concept of PMF is to use the weighted least square fit method
and apply a non-negative constraint with the weight calculated based on analysis uncertainties (Paatero and
Tappert, 1994) (Eq. (S1), Supplement S2). In this study, the input matrix of the PMF comprises 25 chemical
species, including $PM_{10}$ (set as the total variable), carbonaceous fractions (OC*, EC), ions ($Cl^-$, $NO_3^-$, $SO_4^{2-}$, $Na^+$,
$NH_4^+$, $K^+$, $Mg^{2+}$, $Ca^{2+}$), organic tracers (MSA, levoglucosan, mannosan, polyols) and trace metals (As, Ba, Cd,
Cr, Cu, Ni, Pb, Sb, V, Zn). The trace metals were the daily measured metals in some periods (2013, 2017-2018,



2020-2021) and the daily imputed metals. The OC* (=OC minus the sum of the carbon mass from the organic
tracers used in the input variables) was used to avoid considering twice the mass of C atoms in organic markers.
Polyols were calculated as the sum of arabitol and mannitol, supposing that their origin is similar (Samaké et al.,
2019a). The input uncertainties were calculated based on the concentrations and the uncertainties in the analysis
(Gianini et al., 2012; Waked et al., 2014). Details on the calculation of OC* and uncertainties of PMF input are
presented in Section S3, Supplement. The selection of the input variables is guided by our previous yearly PMF
studies at this site (Borlaza et al., 2021; Srivastava et al., 2018; Weber et al., 2019).
**2.5.2. Set of constraints**
The application of PMF constraints is recommended in the European guide on air pollution source apportionment
with receptor models (Belis et al., 2014) to avoid mixing between some factors and reduce the uncertainty of the
rotational ambiguity. The constraints used in this study are also based on the previous PMF studies in Grenoble
(Borlaza et al., 2021; Srivastava et al., 2018; Weber et al., 2019) and are detailed in Table S3.
**2.5.3. Choice of the final PMF solution**
Several solutions, including those from 4 to 11 factors, were investigated to determine the optimal output. This
selection is based on the ratio of $Q_{true}/Q_{robust}$ (evaluating the outlier's effect), the clarity of the chemical profile,
the contribution of factors to $PM_{10}$, the correlation between measured and predicted concentration, and the stability
of the solution. This stability was evaluated using the bootstrapping (BS) and displacement (DISP) methods. BS
analysis randomly resamples the data observation matrix and uses it to run a new PMF. The base-run and boot-
run factors are matched if their correlation exceeds the threshold (generally chosen at 0.6). DISP estimates each
species' uncertainty in the factor profile by fitting the model many times until this variable turns displaced (upper
or lower) from its fitted value. The details of the set criteria for validation are presented in S4.
To evaluate the stability of the PMF solution over time (including possible changes in the chemical profiles of the
sources), we also implemented separated PMF SA for every successive period of 3 years (2013-2016, 2017-2020,
2021-2023) and then we investigated the homogeneity of the chemical profiles by using the Pearson distance (PD)
and standardized identity distance (SID) metrics (Belis et al., 2015). The chemical profiles of PMF solutions every
3 years and 11 years, and those published in Borlaza et al. (2021) are compared to assess the homogeneity of the
chemical profiles.
**2.6. Regression techniques for $PM_{10}$ OP SA**
The regression technique is applied to apportion $OP^v$ (AA, DTT) and PMF-derived $PM_{10}$ sources' contribution, as
expressed in Eq.2. Principally, $OP^v$ (nmol min$^{-1}$ m$^{-3}$) is treated as a dependent variable, and PMF-derived $PM_{10}$
sources' contribution (μg m$^{-3}$) are independent variables. The OP SA methodology in this study follows the
methodology reported by Ngoc Thuy et al. (2024).

$$OP_v = \sum_{p}^{i=1} OP_m^i * PM^i + e \qquad (2)$$

Where:
$OP_v$ is the volume-normalized OP (nmol min$^{-1}$ m$^{-3}$)
$p$ is the number of PMF-derived $PM_{10}$ sources



$OP_m^i$ is the regression slope, denoted as the intrinsic OP of source i (nmol min$^{-1}$ µg$^{-1}$)
$PM^i$ is the contribution of source i to PM$_{10}$ (µg m$^{-3}$)
$e$ is the residual of the regression technique (nmol min$^{-1}$ m$^{-3}$)
The appropriate regression tool is selected based on the collinearity among independent variables and the variance
of regression residuals (Ngoc Thuy et al., 2024). The collinearity among PMF-derived sources was tested using
the variance inflation factor (VIF), which is calculated using Eq. (S3) in Supplement S2 (Craney and Surles, 2002;
O'Brien, 2007; Rosenblad, 2011). The variance of the regression residual is assessed using the Goldfeld-Quandt
test (Goldfeld and Quandt, 1965) to investigate if the regression residual varies by the value of the dependent
variable (OP$^v$). The most appropriate regression method is selected among a wide choice of possible tools
(including ordinary least square, weighted least square, positive least square, Ridge, Lasso, random forest, and
multiple layer perceptron), following the methodology developed by Ngoc Thuy et al. (2024). It is performed with
considering the characteristics of the data and comparing the accuracy metrics (R-square, root mean square error,
and mean absolute error) for each of them. For instance, if the regression residual is constant (homoscedasticity),
the model ordinary least square (OLS) and Positive least square (PLS) are satisfactory. On the other hand, if the
regression residual varied with the dependent variable (heteroscedasticity), the models incorporating some sort of
weighting are chosen (including weighted least squares (WLS) and weighted positive least squares (wPLS)),
where the weighting is the standard deviation of replicated OP analyses.
The most appropriate model was trained by randomly choosing 80% of the dataset and validated with the
remaining 20%. The model was run 500 times to ensure the robustness of the results, especially considering the
remarkable seasonality of many components in the dataset. The contribution to OP of each source is calculated
by multiplying its contribution to PM$_{10}$ with the arithmetic mean intrinsic OP (or regression slope) of the 500
iterations.
**2.7. Seasonal-trend using LOESS decomposition**
Seasonal-trend decomposition using LOESS (SLT) was developed by RB Cleveland et al. (1990) and is a robust
method for decomposing time series into trends, seasonality, and residuals. This method uses LOESS, a method
for estimating the non-linear relationships to decompose a time series. In our case, we used monthly average
concentration as input data in order to have a more robust data set, smoothing high variability noise. The trend
component is first calculated by applying a convolution filter to the data. Then, this trend is removed from the
series. Finally, the average of this detrended in each period is the seasonal component. The residuals can be
explained neither by trend nor by season. The STL is an iterative model that uses LOESS to smooth seasonal and
trend components to obtain the minimum residuals. Further, in STL decomposition, the outliers in the data are
given less weight in the estimation of trend and season. The STL model is described in the equation below:
$$y_t = S_t + T_t + R_t \ (t = 1, 2, \ldots, n) \quad (3)$$
where, in our case, $y_t$ is the monthly contribution of PMF-derived sources, $S_t$ is the seasonal component, $T_t$ is the
trend component, and $R_t$ denotes the residual component. The seasonal frequency was chosen 6 months before
and 6 months after the evaluated month (seasonal frequency = 13 months) to estimate the yearly trend cycle.
Hence, the first and last 6 months of the decomposition time series were removed from the results to prevent edge
effects.



The long-term trend of $PM_{10}$ sources was accessed by applying the STL model to the monthly contribution of
sources to $PM_{10}$ (output of PMF). The fit line of the trend was assessed by using ordinary least squares linear
(OLS). The annual rate change of the trend is the slope of the fit line multiplied by 12 months (µg m$^{-3}$ yr$^{-1}$/ nmol
min$^{-1}$ µg$^{-1}$ yr$^{-1}$). The STL decomposition and the fit line of the trend were performed in Python 3.9 using the
package "statsmodels" (Seabold and Perktold, 2010).
**3. Results and discussion**
**3.1. Evolution of $PM_{10}$ concentration and chemical components**
The annual average concentration of $PM_{10,}$ considering all available daily measurements, is 19.0±10.6 µg m$^{-3}$ for
the whole studied period (2013-2023). The highest annual concentration is observed in 2013 (24.4±13.7 µg m$^{-3}$),
and the lowest is in 2021 (15.3±9.8 µg m$^{-3}$). The number of days with concentrations surpassing the European
standard daily thresholds (40 µg m$^{-3}$) is 176 days in 11 years, representing 4.6% of the total observed days, which
are principally found in the cold season (Nov, Dec, Jan, Feb, Mar).
The $PM_{10}$ main components are organic matter (assuming OM = 1.8*OC (Favez et al., 2010)), representing on
average over the overall period 41.3±8.0% of $PM_{10}$ mass concentration, followed by dust (9.6± 4.4%), nitrate
($NO_3^-$, 7.5±6.2%), non-sea salt sulfate (nss-$SO_4^{2-}$, 7.4±2.4 %), elemental carbon (EC, 5.5±2.5%), ammonium
($NH_4^+$, 3.9±2.0%), sea salt ($Na^+$ and $Cl^-$, 1.7±0.8%) and other non-dust elements (Cu, Pb, V, Zn, representing
0.2±0.1%). These main composition fractions are estimated using the formula as shown in S2, Eq. (S4). The
monthly evolutions of $PM_{10}$ and its main chemical components for the whole period are shown in Figure 2. The
maximum concentration of $PM_{10}$ was observed in winter months (December, January, and February),
corresponding to the highest concentration of OM and EC (7.82±3.11 µg m$^{-3}$ and 1.09±0.74 µg m$^{-3}$, respectively).
Nitrate concentrations are higher in the middle of winter and the early spring, corresponding also with the high
concentrations of ammonium (1.63±1.87 and 0.78±0.62 µg m$^{-3}$). The agricultural activities (especially manure
spreading) could explain this high contribution in spring under humidity and temperature conditions favoring the
condensation of ammonium nitrate in the particulate phase. Nss-sulfate concentrations are more abundant in the
warmer season (summer), where the photochemical production is favorable. No clear seasonal pattern could be
observed for other components (sea salt, dust, non-dust, estimated as described in section S2), suggesting that the
emissions of these components are stable for the whole year. At first glance, decreasing trends appear visible for
$PM_{10}$ and OM, EC, $NO_3^-$, $NH_4^+$, and non-dust components, while sea salt, dust, and nss-$SO_4^{2-}$ do not seem to
present significant trends. With chemical components coming from several emission sources, an advanced
analysis, including a PMF model followed by an STL decomposition, was performed to assess the trend of $PM_{10}$
sources. The result of the PMF model is presented in section 3.2, and the tendencies of $PM_{10}$ sources and OP are
shown in sections 3.3 and 3.4, respectively.





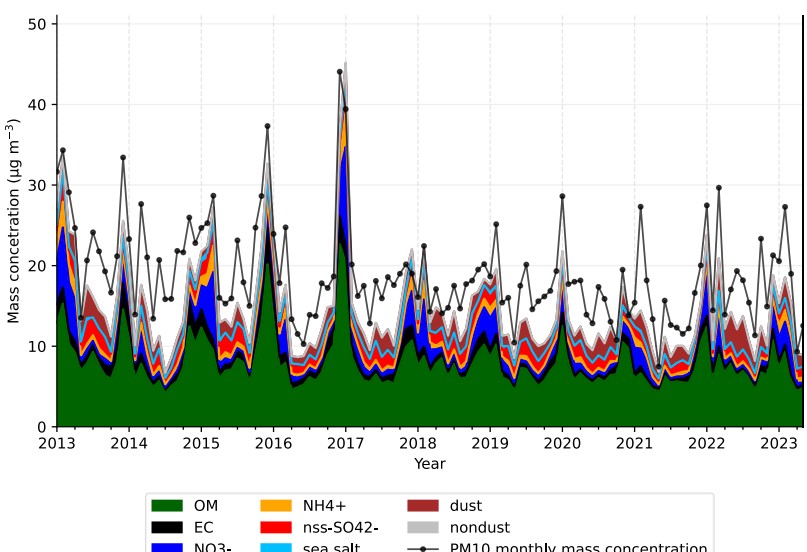

**Figure 2. The average monthly evolution of PM$_{10}$ and its main components from 2013 to 2023. The line represents the**
**monthly average concentration of PM$_{10}$ measured by TEOM-FDMS.**
**3.2. PM$_{10}$ sources apportionment**
**3.2.1. PMF chemical profiles**
Using a unique chemical profile for each of the sources for such a long-term period can potentially limit the
assessment of its evolution (Borlaza et al., 2022). To evaluate such a phenomenon in our case, we investigated
the chemical profile and contribution of PM$_{10}$ sources for three distinct periods (2013-2016, 2017-2021, 2022-
2023) and compared the results with those for the full 11-year period, as well as to the results presented in (Borlaza
et al. (2021) for the year 2017. Particularly, we checked the similarity of the chemical profiles of these PMF
solutions using PD and SID metrics (Belis et al., 2015).
For each SA, the PMF solution was tested from 4 to 11 factors and validated by the criteria presented in section
S4. The results of these validations (Q$_{true}$/Q$_{robust}$, bootstrap run, displacement run, and statistical validation) are
presented in S5, Tables S4, S5 and S6. The runs of 4 to 9 factors returned at least one merging factor, and the
solution with 11 factors led to a factor without geochemical identity. Finally, for each PMF tested (11 years, 2013-
2016, 2017-2021, 2022-2023), the best solution includes 10 PM$_{10}$ sources, with mineral dust, sulfate-rich, primary
traffic, biomass burning, primary biogenic, nitrate-rich, MSA-rich, aged sea salt, industrial, and chloride-rich.
The similarity of the chemical profiles is presented in Figure 3. Most of the factors (i.e., aged sea salt, mineral
dust, primary biogenic, biomass burning, primary traffic, industrial, nitrate-rich, and sulfate-rich) present quite
homogenous chemical profiles over the 3 successive periods, indicating that these source profiles are quite stable
during the full 11-year period and similar compared to sources reported in Borlaza et al. (2021). The MSA-rich
and chloride-rich sources are the most divergent but are still within the limit of the accepted PD and SID range;
however, their standard deviations for PD are slightly higher than for the other sources (Figure 3). This is due to
differences in the contributions of SO$_4^{2-}$ in the chemical profile of MSA-rich, which varied from 6 % to 17%, and
that of Cl$^-$ (73% - 83%) in the chloride-rich factor. In a previous study, Weber et al. (2019) also reported that the



proportion of $SO_4^{2-}$ in the MSA-rich source can significantly vary across French sites, from 6% to 24%. The
chloride-rich source in our study (previously named sea/road salt in Borlaza et al. (2021) is essentially composed
of a high proportion of $Cl^-$, with less than 10% of $Na^+$ and some metals (Cu, Mn, Ni, V). This source is detected
in other alpine valley environments (Glojek et al., 2024), with a similar temporal evolution as here. Since chloride
depletion from the particulate phase can greatly depend on solar radiation, relative humidity, and temperature, the
chemical profile of this factor can vary on different time scales. This source was also observed to be heterogeneous
in the three neighboring sites investigated within 15 km in the previous study in Grenoble (Borlaza et al., 2021).
Nevertheless, it should be noted that it represents only a very minor fraction of the $PM_{10}$ total mass (about 1%).
With these stabilities of the chemical profiles over the years, the solution for the 11-year SA is considered suitable
for further data analyses in this paper. In the next section (3.3.2), we investigate how the contribution of these
sources to total $PM_{10}$ loadings changed over time.

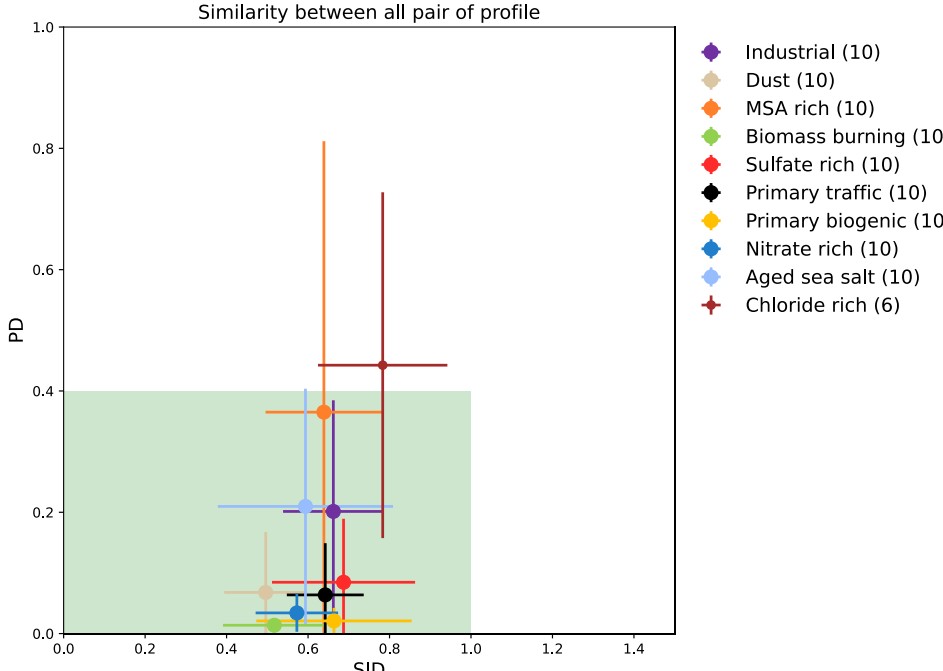

**Figure 3. Similarity plots of the chemical profiles of the solution for the 11-year SA against the 3 SA solutions every 3**
**years, and those presented by Borlaza et al. (2021). The shaded area (in green) shows the limit of the homogeneous**
**chemical profile. For each point, the error bars represent the standard deviation when comparing all pairs of SA**
**solutions (number of pairs in parentheses in the legend).**
**3.2.2. Variations of the source's contribution in the 11-year PMF SA**
As presented in Figure 4, the optimal PMF solution for the 11-year time series identifies 10 $PM_{10}$ sources, with
the contributions of mineral dust (20.9%), sulfate-rich (19.7%), traffic (16.0%), biomass burning (13.5%), primary
biogenic (10.7%), nitrate-rich (7.2%), MSA-rich (6.2%), industrial (2.2%), aged sea salt (2.5 %), and chloride-
rich (1.0%). The chemical profile and contribution of each source are shown in Figures S3 and S4, respectively.
Even though the chemical profiles are homogenous, the contributions of these sources show minor differences
from those reported for this same site by Borlaza et al. (2021) and Srivastava et al. (2018), partly because of the



differences in the respective periods of the studies. However, the main sources are similar, i.e., SIA (nitrate and
sulfate-rich), mineral dust, biomass burning, and primary traffic. Similar general results are also presented for
Swiss Alpine (Ducret-stich and Tsai, 2013), French Alpine (Weber et al., 2018), and Slovenian Alpine areas
(Glojek et al., 2024), showing biomass burning and secondary inorganic aerosols being the most abundant
contributions to PM mass. Primary biogenic and MSA-rich sources are the biogenic sources rarely reported in the
literature; however, they account together for 17% of total $PM_{10}$ mass on average in our study, which is in line
with those reported in urban background sites in France (Samaké et al., 2019b; Weber et al., 2019). The absolute
$PM_{10}$ source contributions are also compared to the average annual concentration of $PM_{10}$ mass to demonstrate
the ability of the PMF model to reconstruct the $PM_{10}$ mass. The difference between observed and reconstructed
$PM_{10}$ concentrations on the 11-year average is about 1 µg m$^{-3}$ (5 %), with no more than 2 µg m$^{-3}$ for any single
year, demonstrating that the PMF model performs well at reconstructing the $PM_{10}$ concentrations.

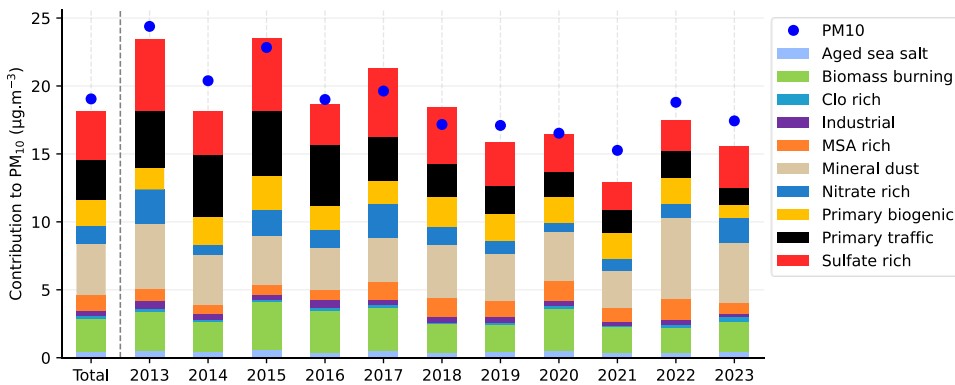

**Figure 4. The absolute average contribution of sources to $PM_{10}$ for every year and the 11 years (total), and the**
**concentration of $PM_{10}$ (blue circle).**
Significant trends in source contributions over this 11-year period are detected (and discussed in section 3.3);
nevertheless, the main contributors to the total $PM_{10}$ mass do not change, with mineral dust, biomass burning,
sulphate-rich, nitrate-rich, and primary traffic being the main contributors to $PM_{10}$. The highest $PM_{10}$
concentrations (observed in winter/spring 2013 and 2015) are associated with the highest contribution of SIA and
biomass burning sources. On the other hand, the relative contribution of SIA and biomass burning showed a
negligible difference (varied from 0.3 to 4%) between these years compared to 2014 and 2016 (Figure S5). The
lowest $PM_{10}$ annual concentration was detected in 2021, notably when the third COVID-19 pandemic lockdown
restrictions applied in France. In addition, the relative contributions (see Figure S5) showed only small changes
compared to those in other years, with an increasing contribution of primary biogenic sources in 2021 (4%
compared to 2020), and only a very light decrease in the anthropogenic sources.
The decrease in $PM_{10}$ annual average concentrations observed since 2017 is associated with decreases in the
contribution of some of the anthropogenic $PM_{10}$ sources. However, using yearly averages for trend analysis may
prevent a proper understanding of the variation in time and of the estimation of the trends based on monthly
averages, which might be more informative, as discussed in section 3.3.



### 3.3. Trends in sources' contributions

#### 3.3.1. Mean rate change in the contribution of $PM_{10}$ sources

The source contribution trend analysis was achieved through STL deconvolution (see section 2.6). These trends for all sources over the full period of the study are presented in Table 1. In this table, the part labeled "Rest" represents the difference between the total $PM_{10}$ measured mass and the sum of the mass of all PMF-derived factors in order to assess any trend of the unresolved part of $PM_{10}$ within our SA study.

$PM_{10}$ concentrations present a downward trend from 2013 to 2023, with an average diminution of 0.73 µg m$^{-3}$ yr$^{-1}$ (3.9%) (S6, Figure S6). Such a downward trend of $PM_{10}$ in Grenoble is in line with that observed in other urban sites in Europe (Aas et al., 2024; Borlaza et al., 2022; Caporale et al., 2021; Colette et al., 2021; Gama et al., 2018; Li et al., 2018; Pandolfi et al., 2016). The reduction of $PM_{10}$ in Grenoble during this period is significantly larger than that in 30 rural sites of the European Monitoring and Evaluation Programme (EMEP) from 2000 to 2017, which show reductions of $PM_{10}$ from -0.008 to -0.58 µg m$^{-3}$ (-1.5% to -2.5%) (Colette et al., 2021). However, the results of our study are highly coherent with results from Aas et al. (2024), presenting a reduction of $PM_{10}$ in 2 rural sites in France (La Tardière and Revin) of -3.5% yr$^{-1}$ between 2005 and 2019. Indeed, France is amongst the EU countries with the highest reduction trend, as presented by Aas et al. (2024).

The anthropogenic sources, such as primary traffic, sulfate-rich, and biomass burning, display the highest decrease between 2013 and 2023 in Grenoble, with a reduction of 0.37, 0.25, and 0.13 µg m$^{-3}$ yr$^{-1}$ (12.9, 6.9, and 5.5%), respectively. The other anthropogenic sources also have a significant decreasing trend; however, they are much lower (nitrate-rich: -0.11 µg m$^{-3}$ yr$^{-1}$, industrial: -0.02 µg m$^{-3}$ yr$^{-1}$). The downward trends of these anthropogenic sources (mainly traffic, SIA, and industrial) were also underlined for other European urban sites (Colette et al., 2021; Diapouli et al., 2017; Pandolfi et al., 2016). For instance, a similar approach was followed by Pandolfi et al. (2016), investigating the Mann-Kendall trend of PMF-derived sources, and reported an almost equivalent downward trend of the sulfate-rich factor of -0.32 µg m$^{-3}$ yr$^{-1}$ between 2004 and 2014 in Spain. The decreasing trends of primary traffic, domestic biomass burning, and industrial emissions are potentially influenced by the reduction in primary emissions due to various abatement strategies (as discussed in the following subsections, notably in 3.3.3 and 3.3.4).

Conversely, natural sources such as mineral dust and chloride-rich factors do not show any significant trend or follow a very weak one (aged sea salt, primary biogenic). MSA-rich is the only source that displays a significant upward trend, with an increase of 0.08 µg m$^{-3}$ yr$^{-1}$; further studies would be needed to relate this last increase to changes in precursor emissions or reactivity during transport. Finally, the low evolutions in the contributions of the natural sources demonstrate that the reduction in $PM_{10}$ in Grenoble is essentially related to the reduction of anthropogenic activities, especially sources related to traffic and domestic biomass burning activities.

**Table 1. Trend of $PM_{10}$ sources and $PM_{10}$ (in µg m$^{-3}$ yr$^{-1}$ and % yr$^{-1}$).**

|  | Absolute trend (µg m$^{-3}$ yr$^{-1}$) | Relative trend (% yr$^{-1}$) | P-values | $R^2$ |
|---|---|---|---|---|
| Aged sea salt | -0.01 | -2.50 | <<0.01 | 0.22 |
| Biomass burning | -0.13 | -5.48 | <<0.01 | 0.98 |
| Chloride rich | 0.00 | 1.18 | 0.01 | 0.07 |
| Industrial | -0.02 | -5.36 | <<0.01 | 0.40 |
| MSA rich | 0.08 | 6.63 | <<0.01 | 0.64 |



| | | | | |
|---|---|---|---|---|
| Mineral dust | 0.04 | 1.03 | 0.02 | 0.05 |
| Nitrate rich | -0.11 | -8.08 | <<0.01 | 0.94 |
| Primary biogenic | -0.01 | -0.49 | 0.03 | 0.04 |
| Primary traffic | -0.37 | -12.85 | <<0.01 | 0.94 |
| Sulfate rich | -0.25 | -6.89 | <<0.01 | 0.70 |
| $PM_{10}$ | -0.73 | -3.89 | <<0.01 | 0.68 |
| Rest | -0.11 | -2.13 | <<0.01 | 0.39 |

420

### 3.3.2. Potential influence of meteorology

The STL deconvolution is inherently constructed to separate the yearly and seasonal variations from the long-term trends. While we discuss the long-term trends of the sources in other sections (3.3.1, 3.3.3, and 3.3.4), it is also interesting to evaluate the impact of the meteorology on the seasonal variations of the concentrations. It is well known that inversion layers in the lower atmosphere are extremely important for the modulation of the concentrations at the ground, particularly in the context of Alpine valleys during winter (Carbone et al., 2010; Glojek et al., 2022). In this section, we tried to better evaluate these impacts on the concentrations from the sources of PM in the case of our time series.

This was considered with the measurements of temperature along the slopes of the mountains very close to the city center (as described in section 2.2.3), for the winter periods of 2017-2023. It has been previously shown by Allard et al. (2019) that such measurements are representative of the temperature in the valley, despite the potential influence of wind slopes. We particularly considered the temperature gradient over the first 700 m above ground and the number of days with persistent inversion, as defined in section 2.2.3.

The analysis of the relationship between the $PM_{10}$ and bulk temperature vertical gradients ($\Delta T / \Delta z$) in winter (Nov, Dec, Jan, Feb, Mar), summer (May, June, Jul, Aug), and transition season (remaining months) reveals that thermal inversion events and high $PM_{10}$ concentration are mainly occurring in winter time (Supplement S7, Figure S8) during the 5 years of the study. Periods of persistent temperature inversion were assessed based on the condition in Eq. 1, which detected 79 persistent inversion days in series from 4 to 22 consecutive days, for the winter periods 2017-2023. A meaningful correlation is obtained between $PM_{10}$ concentrations and bulk temperature vertical gradient (r reaching 0.60, p<<0.001) for these winter months and even better when considering only the persistent inversion periods (r reaching 0.67, p<<0.001) for individual years (Table S7).

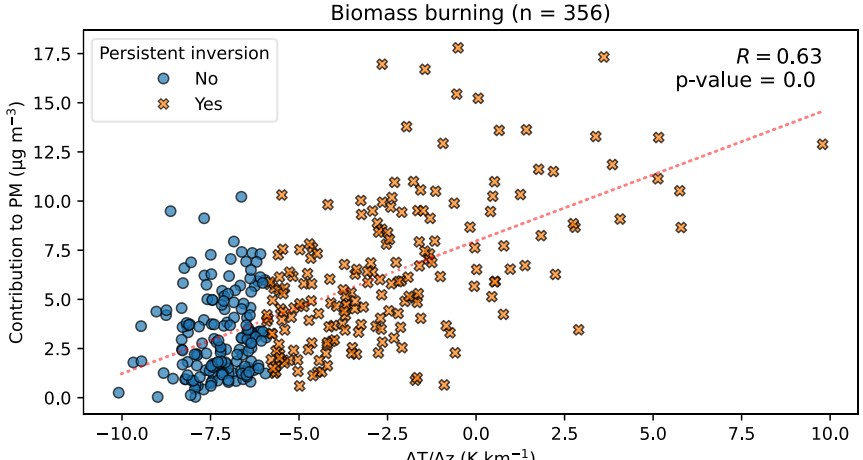

**Figure 5. Daily concentrations of biomass burning to PM$_{10}$ and daily temperature gradients ($\Delta T/\Delta z$) during the winter periods (from November to March) of 2017-2023. The dotted red line is the linear regression fit. The blue circle symbols denote days when persistent inversion does not occur, and the orange multiple symbol denotes days when persistent inversion occurs.**

The distribution between the daily PM$_{10}$ concentration and daily average $\Delta T/\Delta z$ in winter months revealed that the majority of PM$_{10}$ concentration peaks (in excess of 40 µg m$^{-3}$) occur during the persistent inversion days (Figure S9). However, it also shows that a few high PM$_{10}$ concentrations could be found on the days without persistent inversion; meanwhile, the days with persistent inversion do not always have high PM$_{10}$ concentrations. This result is not surprising since the concentration of PM$_{10}$ is not only associated with thermal inversion events but also depends on other meteorological conditions (precipitation, heat deficit) and the variation of pollutant emissions (Carbone et al., 2010; Largeron and Staquet, 2016).

Interestingly, the impact of persistent inversion days on PM$_{10}$ concentrations from the residential biomass burning source is larger than that for other sources or total PM$_{10}$ (Figure 5), with a higher correlation (0.63). In addition, the contribution of this source is systematically lower during non-inversion days, and large concentrations are essentially made during persistent days. The large impact of the inversions on the local sources is confirmed when comparing the source contribution of the inversion days vs non-inversion days (Figure 6). This figure shows both the large increase in average PM$_{10}$ concentrations and also the contributions of the local sources (emissions from residential biomass burning, traffic, industries, mineral dust probably from resuspension) in the cases of inversion days during winter. Conversely, long-range transport sources (sulfate-rich, nitrate-rich) tend to be less important during these inversion days. A similar pattern is observed for the relative contribution of sources to PM (Figure S.10), in which the significant contribution of biomass burning, dust, industrial, and primary traffic is detected during inversion events. The trends of the two most important local anthropogenic sources (domestic biomass burning and traffic) are further discussed in the next sections.





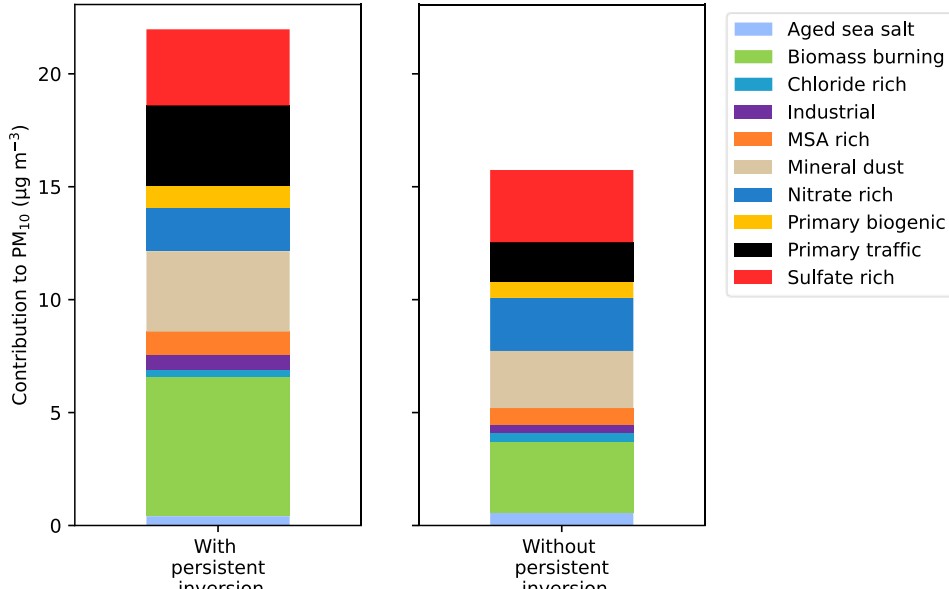

**Figure 6. Contribution of the different sources to the PM₁₀ composition for days with persistent inversion vs non-**
**inversion days of the winters 2017-2023.**
**3.3.3. Trend in biomass burning contributions**
The trend of the domestic biomass burning $PM_{10}$ concentrations is investigated via an STL decomposition analysis
on this PMF-derived source (Figure 7), indicating a statistically significant decreasing trend from 2013 to 2023
(p-values $\ll 0.01$). The seasonal estimate shows the highest values in the winter season (Nov, Dec, Jan), with a
visual trend to a smoothing of the peak concentrations; conversely, from Mar to Sept, the seasonal variations
showed constantly lowest values. Extreme residual values were detected in the winter months of 2016, 2017, and
2021, explained by high-concentration episodes of $PM_{10}$, where the concentration exceeded the European standard
for $PM_{10}$ concentration in 24 hours ($PM_{10}$ concentration varied from 50 to 78 µg m$^{-3}$). The linear fit line of the
trend is highly significant with $R^2 = 0.97$, with a reduction of 134 ng m$^{-3}$ yr$^{-1}$ (-5.5% yr$^{-1}$).





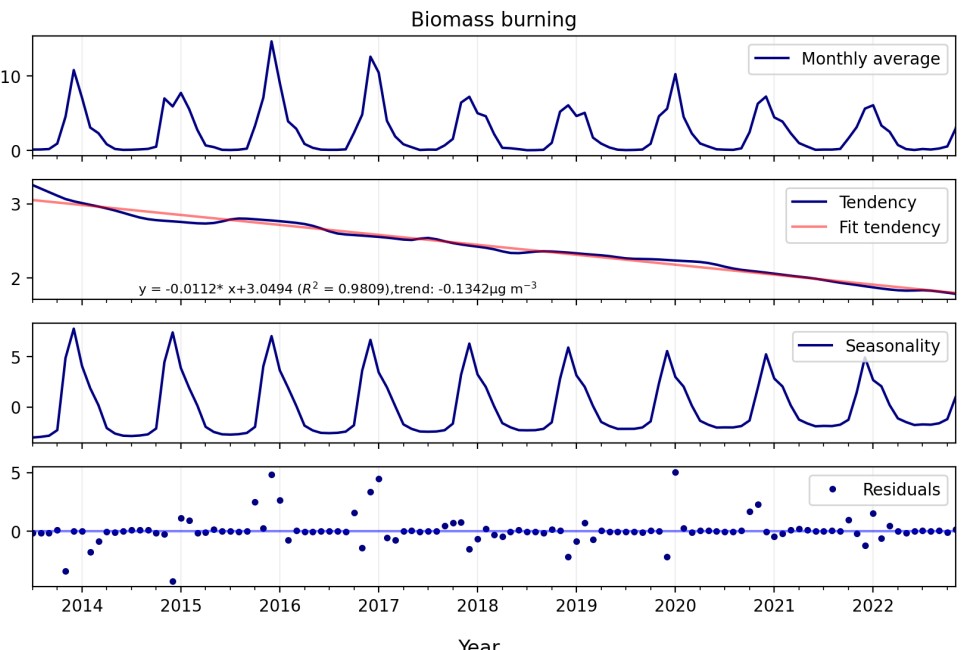

**Figure 7. The season-trend (STL) decomposition of biomass burning**
This reduction of biomass burning concentrations in Grenoble is 4 times higher than the results from a long-term
study (2012 to 2020) in a French rural site - (Observatoire Pérenne de l'Environnement, OPE) (Borlaza et al.,
2022) - estimated at 33 ng m$^{-3}$ yr$^{-1}$ over the same period. Besides the study of Borlaza et al. (2022), there are no
previous PMF studies describing any trend of biomass burning factors. Nevertheless, similar trends were found
for concentrations of biomass burning tracers. In particular, Font et al. (2022) presented a downward trend of
$PM_{10}$ concentration from wood burning (a reduction from 1.5 to 3.8 % yr$^{-1}$ ) in urban sites in the United Kingdom
from 2010 to 2021, by calculating the emission of wood burning from aethalometer measurement. Similarly, from
2002 to 2018 in Norway, a downward trend of 2.8% yr$^{-1}$ was also detected for levoglucosan (Espen Yttri et al.,
2021). Additionally, Colette et al. (2021) modeled the trend of the emissions from different activities in Europe,
showing that the trend of $PM_{10}$ heating emissions was decreasing in the period 2000-2017, with mean rate values
varying from 0.8 to 3.3% yr$^{-1}$ for 30 European countries (EMEP monitoring sites). Even though the chemicals
and the period of these studies differ, a decreasing trend is generally observed among European cities, including
the one investigated here. Interestingly, the biomass burning source in Grenoble shows the strongest decreasing
trend, with a reduction of 5.5% yr$^{-1}$.
Since the biomass burning sources in Grenoble are related to residential heating, the observed reduction of the
concentrations from this source could be linked to household behaviors (including appliance renovation) on top
of the changes in meteorological conditions, lowering the overall heating demand. The average annual biomass
burning sources PMF-derived is compared to the local $PM_{10}$ emission inventory for residential heating (tonnes)
in the Grenoble metropolis, estimated by the regional air quality monitoring agency (Atmo AuRA), to confirm
the trend of biomass burning (Figure 8). This emission inventory has been available until 2022.



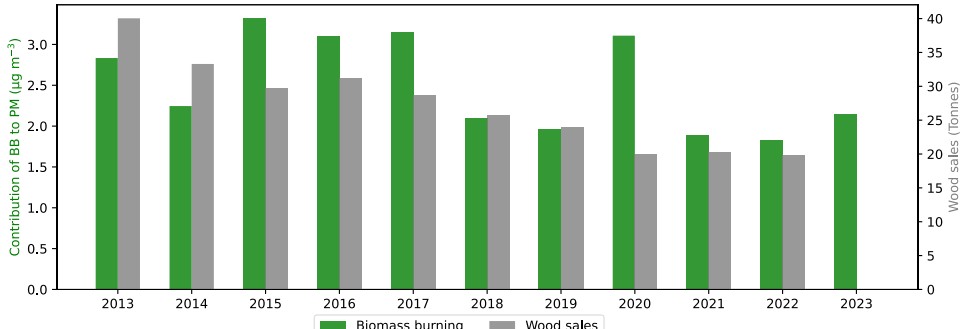

**Figure 8. Comparison between annual average PM$_{10}$ emission inventory based on the quantity of wood sales (in grey) in the Grenoble metropolis and the yearly average PM$_{10}$ concentrations from the PMF biomass burning factor (in green).**

Except for the year 2020, the annual average of biomass burning agreed with the emission inventory, demonstrating the consistency between the sources observed by the PMF model and the local inventory emission data. Since 2015, the Grenoble metropolis has set up an air-wood bonus to encourage households to renew their individual wood-burning appliance (fireplace or stove). It aims to replace all open fireplaces with closed appliances in October 2024. The downward trend of biomass burning concentration could then be considered as partly due to the implementation of dedicated action plans at the regional scale.

### 3.3.4. Trends in traffic exhaust emissions

Similar to the time series of biomass burning concentrations, the traffic contribution was subjected to specific STL analysis (Figure 9). A significant downward trend of the concentrations of PM from traffic emission is detected with a reduction of 374 ng m$^{-3}$ yr$^{-1}$ (12.9% yr$^{-1}$) (p-value << 0.01). This reduction is almost 3 times larger than that of the biomass burning concentrations. Traffic concentration before 2017 also showed a clear seasonality with maxima in winter, which nearly disappeared from 2018 onward. It is striking that the same behaviors (strong downward trend and smoothing) are also observed for NO$_x$ concentration, another indicator of traffic exhaust emission, which is also observed for NO$_x$ seasonal patterns (see Supplement S6 and Figure S7). Residuals show extreme values in the same month as biomass burning in 2016 and 2017, matching the PM$_{10}$ episode. The traffic trend closely follows a linear regression fit line, with R$^2$ = 0.94.



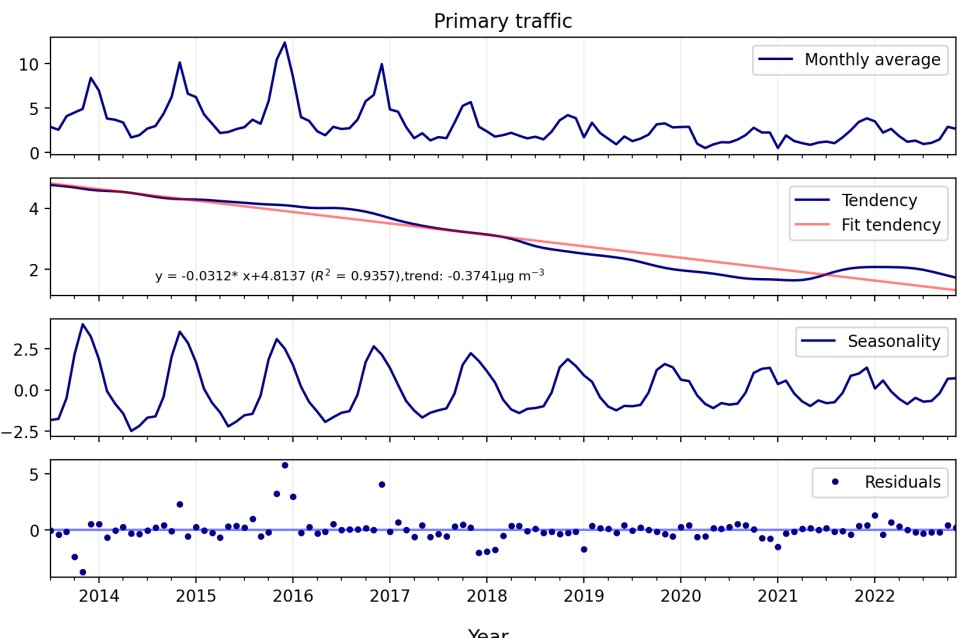

**Figure 9. The season-trend (STL) decomposition of PMF-derived traffic source**
The downward traffic trend observed in this study is consistent with another long-term study (2012-2020) of a
rural site in France, which showed a traffic trend of -0.1 µg m$^{-3}$ yr$^{-1}$ (Borlaza et al., 2022). This is aligned with
other results of fossil fuel black carbon in several rural sites in France (Font et al., 2025), or EC over many rural
sites in Europe (Aas et al., 2024). Additionally, our result also agrees with other studies, like that by Pandolfi et
al. (2016), which indicated a downward trend of traffic sources in an urban site in Spain, with a reduction of 0.11
µg m$^{-3}$ yr$^{-1}$, which is lower than that of our study. Finally, the trend of traffic emission to PM$_{10}$ in 30 European
countries was modeled as reported by Colette et al. (2021), showing a downward trend with a reduction from 2.3
to 3.5% yr$^{-1}$ from 2000 to 2017. As for biomass burning, the Grenoble supersite seems then experiencing faster
reductions in primary traffic PM loadings than most of others European cities.
Furthermore, the PMF-derived traffic factor was compared to the local PM$_{10}$ traffic emission inventory by fuel
type (provided by Atmo AuRA), revealing very similar trends (Figure 10). In addition, this source is also
compared to the PM$_{10}$ emission by the transport sector (kilotonnes) over France, which was assessed from the
emission inventory data of CITEPA (Figure S11), also confirming the concomitant reductions of traffic emissions
and contributions to PM$_{10}$ in ambient air.




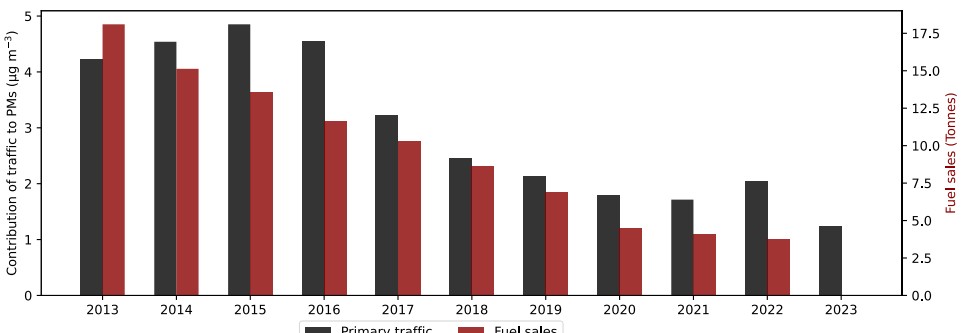

**Figure 10. Comparison between annual average PM$_{10}$ emission inventory based on the quantity of fuel sale (red bar)**
**in the Grenoble metropolis and the yearly average PM$_{10}$ concentrations from the PMF-derived traffic source**
**contributions (black bar).**
This traffic trend may be separated into three parts. Between 2014 and 2016 with a slow decrease trend of -3% yr$^-$
$^1$; from 2016 to 2021, with an average reduction of 10% yr$^{-1}$, and a mild increasing trend of approximately 3% yr$^-$
$^1$ in the last three years of the study. The beginning of this increase coincides with the post-lockdown period, when
transportation activities were back to normal, resulting in a fairly similar contribution of traffic sources compared
to that in the pre-lockdown period.
Besides the implementation of the two versions of the Euro 6 emission standards (introduced in 2015 and 2018,
respectively), local emission abatement strategies decided by Grenoble municipality from 2016 onwards might be
the main drivers for the observed decreasing trends (City's low emission zone
https://zfe.grenoblealpesmetropole.fr/ last assessed: 21/05/2025).
**3.4. Trends in PM$_{10}$ OP sources**
In this section, the sources of OP are assessed using regression techniques, which are presented in section 2.6.
The most appropriate model is selected based on characteristics of PMF-derived sources and OP$_v$, as shown in
section 3.4.1. Intrinsic OP derived from the best regression model, indicating the highest redox-active PM sources,
is presented in section 3.4.2. Finally, section 3.4.3 provides the trend of OP sources, highlighting which sources
are the drivers of OP trends.
**3.4.1. Selection of the most appropriate model**
Following the methodology exposed in Ngoc Thuy et al. (2024), the characteristics of the dataset, including
collinearity and heteroscedasticity, are tested in order to select a satisfactory inversion model for OP$_{DTT}$ source
apportionment (SA) and OP$_{AA}$ SA (Table S8). The OP SA can be applied for the 11-year PMF solution since the
source profiles have been demonstrated to be homogenous over the years. Consequently, the OP$^m$ should be
substantially homogenous over the years (Ngoc Thuy et al., 2024), and it is unnecessary to perform the OP SA
for each year separately. The characteristic tests indicate that the weighted positive least squares (wPLS) and
weighted least squares (WLS) could be suitable models for both OP$_{AA}$ and OP$_{DTT}$ SA. The average accuracy
metrics of the testing dataset in 500 iteration runs (including R$^2$, RMSE, MAE) of wPLS and WLS were compared
to select the most appropriate model (Table S9). Finally, WLS was chosen due to the highest R$^2$ and lowest error
for both OP$_{AA}$ and OP$_{DTT}$ prediction. The comparison between observed and predicted OP$_{AA}$ and OP$_{DTT}$ showed



a good correlation between measured OP and WLS predicted OP, with $R^2 = 0.80$ and 0.70 for $OP_{AA}$ and $OP_{DTT}$,
respectively (Figure S12 and S13), with n = 1570 for $OP_{AA}$ and $OP_{DTT}$.
In addition, the study revealed good performance of MLP and RF for the training and testing datasets (Table S10).
These neural network models were overfitting the results of OP SA for the 6 French sites tested in Ngoc Thuy et
al. (2024) since the number of samples was lower than 200 for individual sites. The present study confirmed the
conclusion of Ngoc Thuy et al. (2024), demonstrating that a higher number of samples improved the performance
of the neural network model. However, such non-linear models do not provide values for intrinsic OP, and cannot
be selected for the final results at this stage.
**3.4.2. Intrinsic OP of PMF-derived sources**
The intrinsic OP of 1µg $PM_{10}$ source ($OP^m$ nmol min$^{-1}$ µg$^{-1}$) is investigated thanks to the WLS technique, resulting
in 500 values of $OP^m$ for each source (Table 2 and Table S11). The anthropogenic sources, including biomass
burning, industrial, and traffic, have the dominant intrinsic $OP_{DTT}$ and $OP_{AA}$, which is consistent with the study in
2017-2018 in Grenoble (Borlaza, 2021) and results obtained at other French sites (Ngoc Thuy et al., 2024; Weber
et al., 2021) and EU sites (Fadel et al., 2023; Veld et al., 2023). The different ranking of the intrinsic OP of the
sources according to the two assays is also aligned with previous results (Weber et al., 2021). While intrinsic
$OP_{AA}$ of biomass burning is highest (0.76 nmol min$^{-1}$ µg$^{-1}$), followed by industrial (0.48 nmol min$^{-1}$ µg$^{-1}$) and
traffic (0.38 nmol min$^{-1}$ µg$^{-1}$), the order of intrinsic $OP_{DTT}$ is industrial (0.52 nmol min$^{-1}$ µg$^{-1}$), traffic (0.38 nmol
min$^{-1}$ µg$^{-1}$) and biomass burning (0.14 nmol min$^{-1}$ µg$^{-1}$). The intrinsic $OP_{DTT}$ of biomass burning is also lower than
that of $OP_{AA}$, as reported by Borlaza et al. (Borlaza et al., 2021), suggesting the synergistic and antagonistic effects
between some elements, quinones, or bioaerosols, decreasing the overall intrinsic $OP_{DTT}$ of this source
(Pietrogrande et al., 2022; Samake et al., 2017; Xiong et al., 2017).
The other anthropogenic sources, including nitrate-rich and sulfate-rich, have lower intrinsic OP than
anthropogenic sources associated with combustion (traffic and biomass burning), as reported by Daellenbach et
al. (2020). The natural sources have a negligible intrinsic OP (lower than 0.03 nmol min$^{-1}$ µg$^{-1}$). These findings
highlight the high impact of the anthropogenic sources, verified for the overall period 2013-2023.
**Table 2. Intrinsic OPAA and OPDTT (nmol min$^{-1}$ µg$^{-1}$) of PM$_{10}$ sources (mean ± std of 500 iterations)**

| Source | $OP_{AA}$ | $OP_{DTT}$ |
|---|---|---|
| Aged sea salt | -0.02 ± 0.07 | 0.03 ± 0.02 |
| Biomass burning | 0.76 ± 0.13 | 0.14 ± 0.09 |
| Chloride rich | -0.07 ± 0.09 | 0.01 ± 0.02 |
| Industrial | 0.48 ± 0.14 | 0.52 ± 0.08 |
| MSA rich | 0.20 ± 0.04 | 0.01 ± 0.02 |
| Mineral dust | -0.03 ± 0.06 | 0.01 ± 0.02 |
| Nitrate rich | 0.09 ± 0.16 | 0.11 ± 0.12 |
| Primary biogenic | 0.00 ± 0.04 | 0.02 ± 0.03 |
| Primary traffic | 0.38 ± 0.10 | 0.24 ± 0.07 |
| Sulfate rich | -0.01 ± 0.08 | 0.09 ± 0.04 |



### 3.4.3. Trends in OP

The trend of OP is first presented by the yearly average contribution of sources to $OP_{AA}$ and $OP_{DTT}$ (Figure 4), indicating a reduction of OP values over the years. Overall, the yearly average of the $OP_{AA}^v$ and $OP_{DTT}^v$ is decreasing and reached its lowest values in 2021 (2.41 and 1.17 nmol min$^{-1}$ m$^{-3}$ for $OP_{AA}$ and $OP_{DTT}$, respectively). From 2018 onward, both assays consistently exhibited lower $OP^v$ values than in preceding years. Although $OP^v$ is normalized to $PM_{10}$ mass concentration, implying that a decrease in $PM_{10}$ concentration generally reduces OPv, the contribution of sources to OP is different from that of $PM_{10}$. While dust and sulfate-rich are dominantly contribute to $PM_{10}$, biomass burning is the most important contributor to $OP_{AA}$ (1.87 ± 2.7 nmol min$^{-1}$ m$^{-3}$), and primary traffic is commonly assessed as the largest contributor to $OP_{DTT}$ (0.71 ± 0.70 nmol min$^{-1}$ m$^{-3}$). The industrial mass contribution is 10 times lower than that of the sulfate-rich. However, industrial emissions appear to contribute much more to $OP_{AA}$ and equally to $OP_{DTT}$ than the sulfate-rich factor. This finding was also observed in 2017-2018 at the same site in Grenoble (Borlaza, 2021). This significant contribution of traffic and biomass burning over the years is more evident when considering relative contribution (Figure S15). These results again emphasize the importance of considering not only the mass concentration but also its redox activity in evaluating the potential adverse health effects of a source of PM.

In addition, the temporal evolution of $OP_{AA}$ and $OP_{DTT}$ did not exactly follow PM10 trends, especially for the period of 2016-2017 and 2019-2020. Regarding the period between 2016 and 2017, a dramatic increase in $PM_{10}$ concentration is observed, principally due to the higher contribution of nitrate and sulfate-rich. On the other hand, $OP_{AA}$ and $OP_{DTT}$ values remained fairly unchanged between 2016 and 2017. Focus on 2019 and 2020, the PM concentration and $OP_v$ values are identical (less than 0.001 µg m$^{-3}$ and nmol min$^{-1}$ m$^{-3}$ of difference, respectively), while $OP_{AA}^v$ presents a remarkable difference (Δ = 0.8 nmol min$^{-1}$ m$^{-3}$). Indeed, the discrepancy between 2019 and 2020 in $OP_{AA}^v$ is principally attributable to a higher contribution to biomass burning, which is the dominant driver of $OP_{AA}^v$. Overall, the downward trend of $OP_{AA}$ and $OP_{DTT}$ is different from $PM_{10}$, since the driven sources of OP and PM are different.

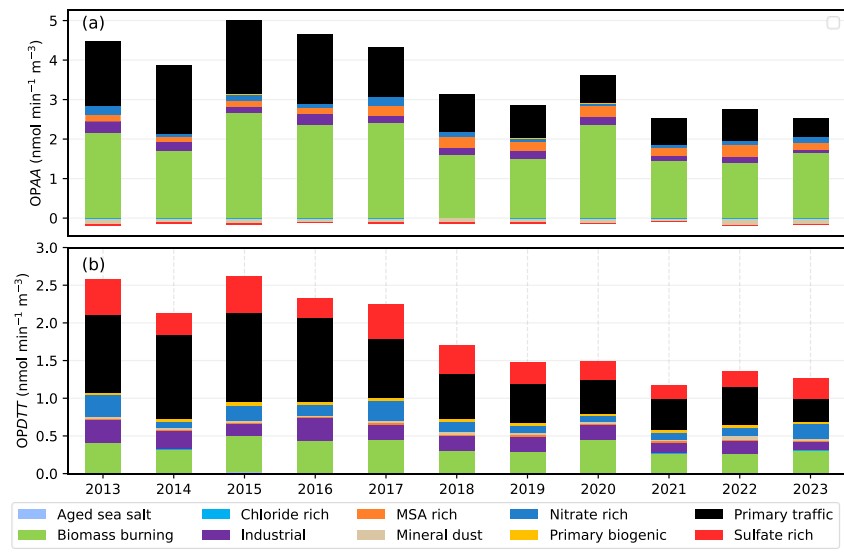

**Figure 11. Yearly average contribution of sources to (a) $OP_{AA}^v$ and (b) $OP_{DTT}^v$**



The yearly average may not be properly representative of the trends of OP; therefore, a STL deconvolution was
performed for $OP_{AA}^m$ and $OP_{DTT}^m$ (Figures S16, S17, respectively) to investigate the trend of $OP^m$ over the 11
years of the study. Indeed, considering the trend of the intrinsic $OP^m$ confirms that the downward trend of some
sources leads to a change in the trend of $OP_{AA}^m$ and $OP_{DTT}^m$.
An insignificant linear trend is observed for $OP_{AA}^m$ (fit line: $R^2 = 0.4$, p-values $\ll 0.01$), yet its average intrinsic
activity still exhibits a decreasing value, with the annual mean falling by approximately 0.002 nmol min$^{-1}$ m$^{-3}$
(2.5 %) across the study period. Interestingly, the seasonality of $OP_{AA}^m$ exactly matches the seasonality of
biomass-burning concentrations, pointing out that the high values of $OP_{AA}^m$ in winter align with biomass-burning
activities. The trend line of $OP_{AA}^m$ did not match the trend of biomass burning nor that of the traffic or industrial
emissions, suggesting the synergistic effect between sources, as well as the influence of the other sources outside
of the winter season, such as MSA-rich and primary biogenic, which get a high ranking of $OP_{AA}^m$ (Table 2).
Conversely, the $OP_{DTT}^m$ showed a significant downward trend ($R^2 = 0.6$, p-value $\ll 0.01$), with a reduction of 0.005
nmol min$^{-1}$ µg$^{-1}$ (6.5%) across 11 years. The seasonality of $OP_{DTT}$ is different from that of biomass burning and
$OP_{AA}^m$, since biomass burning is not the main driver of $OP_{DTT}$ (only ranked third), indicating a lower influence of
this source on $OP_{DTT}^m$ compared to $OP_{AA}^m$. Interestingly, a slight increase in $OP_{DTT}^m$ from 2021 onward is also
observed, which is associated with $PM_{10}$ and traffic, suggesting that traffic emission could be the main driver for
increasing $PM_{10}$ concentration and $OP_{DTT}^m$ from 2021. Overall, the relative decrease of $OP_{DTT}^m$ exceeds that of
$OP_{AA}^m$ could be explained by the 4$^{th}$ most important contributor to these OPs. All four leading contributors to
$OP_{DTT}^m$ show significant reductions, whereas MSA-rich factor, one of the top four contributors to $OP_{AA}^m$, has an
increasing trend. These findings again underscore that trends in $OP^m$ are governed by the evolution of the sources
most active in each assay. Thus, the decrease in the magnitude of the $OP_m$ depends on how its dominant redox-
active sources evolve over time.
Considering the volume-based metrics ($OP_v$), a downward trend is detected for $OP_{AA}$ and $OP_{DTT}$. $PM_{10}$ decreased
by 3.9 % over the decade, which is consistently comparable to $OP_{AA}^v$ (4.9 %) and $OP_{DTT}^v$ (5.3 %). This good
agreement partially reflects the influence of the PM mass concentration since these $OP^v$ values are normalized to
$PM_{10}$ mass concentration. However, the slight difference in the relative downward trend could be related to the
most driven sources of OP and PM, as discussed above.
Finally, the impact of persistent inversion days on the $OP^v$ is also investigated to assess the association between
the redox activity of PM sources and thermal inversion. A comparison of the source's contribution to $OP^v$ (for
both AA and DTT) between the period with and without persistent inversions is carried out and shown in Figure
S14. The comparison confirms the larger increases in average $OP_{AA}$ (85.1%) and $OP_{DTT}$ (63.8 %) compared to
that of PM10 (39.6 %) for the persistent inversion periods. The higher values of $OP_{AA}$ and $OP_{DTT}$ are related to the
larger increases in the contribution of local anthropogenic sources, with BB impacting most the $OP_{AA}$ values while
traffic significantly influences $OP_{DTT}$. This result again highlights the potential effect of persistent inversion on
the PM10 source's contribution, but all the more of their redox-active properties, which could be associated with
the health-relevant metrics (Tassel et al., 2025 in progress).
Over the decade, anthropogenic sources have driven OP, with biomass burning impacting $OP_{AA}$ and traffic/
industrial sources dominating $OP_{DTT}$. Frequent thermal inversion in Alpine valley strongly amplifies OP, which is
more significant than the mass of $PM_{10}$ itself. Finally, $OP_v$ and intrinsic OP trends over the decade do not align



with that of PM$_{10}$ mass, emphasizing the need to prioritize redox-active components over the bulk PM
concentration in air quality policy.

**4. Conclusions**

Thanks to long-term PM$_{10}$ observations with a detailed set of chemical markers, a comprehensive source
apportionment was performed to identify the evolution of PM$_{10}$ sources in Grenoble (France). This is one of the
very few studies in Europe that could assess over 11 years of PM$_{10}$ sources and the only study so far investigating
trends in PM$_{10}$-related OP. The trend of PM$_{10}$ sources, especially anthropogenic sources such as biomass burning
and primary traffic, was evaluated and linked to the meteorology and emission reduction policies. In addition, the
trend of OP$^m$, OP$^v$, and sources of OP revealed that the trend of OP depends on the source that drives OP. The
analysis of these trends confirms the improvement of the air quality at the Grenoble supersite from 2013 to 2023,
and objectivates the main sources that are involved in their concentration' decrease.
Nevertheless, the following methodological limitations in this long-term study shall be kept in mind:
- Daily concentrations of metal elements were only analyzed for some periods (2013, 2017-2018, 2020-2021),
while the remaining data were derived from weekly sampling. An imputation technique was implemented to
impute daily concentrations. The PMF result demonstrated the stability of most chemical profiles at Grenoble
from 2013 to 2023, compared to those previously published (Borlaza et al., 2021), despite these uncertainties in
the imputed metal concentrations.
- The process of implementing such a PMF analysis strategy is not straightforward. A combined PMF approach
could be used for datasets with different time resolution (Via et al., 2023). This approach would allow combining
the 7-day and daily filter samples into a PMF without performing imputation.
- The lack of a secondary biogenic organic aerosol tracer in long-term observations prevents the identification of
the BSOA source, which could make up about 10% of the total mass of PM$_{10}$ on a yearly average, as observed in
previous work at the site (Borlaza et al., 2021), which used 3-MBTCA and picnic acid for the yearly period of
observation.
Overall, a total of ten sources were identified, including aged sea salt, biomass burning, chloride-rich mineral
dust, MSA-rich, nitrate-rich, industrial, primary biogenic, and primary traffic. The source chemical profiles are
consistent with those presented in 2017-2018 (Borlaza et al., 2021), demonstrating that the sources of PM$_{10}$ in
Grenoble were relatively stable during our study period. The trend of PM$_{10}$ sources was investigated using STL
decomposition, which reveals a downward trend for all the PM$_{10}$ sources over 11 years, especially for the
anthropogenic sources. Extending PMF outputs to oxidative potential apportionment showed that biomass
burning, traffic, and industrial emissions dominate redox activity in both the ascorbic acid (AA) and dithiothreitol
(DTT) assays. Trend analysis of volume- and mass-normalized OP metrics indicates that biomass burning governs
the long-term behavior of OP$_{AA}$. In contrast, traffic is the principal driver of OP$_{DTT}$ assay, underscoring source-
specific control of PM$_{10}$ OP in the Grenoble atmosphere.
Both of these anthropogenic sources, as well as their influences on PM$_{10}$ OP, showed significant decreasing trends
concomitantly to the implementation of emission reduction strategies (both at the national and regional levels)
that should be reinforced to reach the goals of the European zero pollution action plan and the recently revised
Directive on ambient air quality (22024/2881/EU). The continuation of these measurements will take place in the
coming years, with this site being selected as one of the supersites for the new EU Air Quality directive.



**Data availability**

The datasets could be made available upon request by contacting the corresponding authors.

**Author contributions**

VDNT performed the source apportionment and the trend of sources, and the result visualisation. GU, JLJ mentoring, supervision, validation of the methodology and results. RE, SD, CV, and AN contributed to data acquisition (analytical investigation on samples) and data curation. OF, JLJ, and GU acquired funding for the original PM sampling and analysis. VDNT wrote the original draft. All authors reviewed and edited the manuscript.

**Competing interests**

The authors declare that they have no conflict of interest.

**Acknowledgment**

The authors would like to express their sincere gratitude to many people of the Air-O-Sol analytical platform at IGE for sample management and chemical analyses. We gratefully acknowledge the personnel at Atmo AuRA (C. Bret, C. Chabanis) for their support in conducting the dedicated sample collection and providing weekly metals data.

**Financial support**

This study was partially funded by the French Ministry of Environment through its contributions to the CARA program. Part of the project was also funded by Atmo AuRA (ensuring filter sampling and costs related to the analyses of the elemental concentrations in the weekly samples. IGE contributed financially to the analyses of ions. The extended analyses for the daily trace elements were funded by the QAMECS program from Ademe (1662C0029).



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
