# Peer review of "Decadal trends (2013-2023) in PM10 sources and oxidative potential at a European urban supersite (Grenoble, France)"

_EGUsphere, 2025_

## Referee Comment (RC2)

This study investigates the trends of PM10 sources and oxidative potential (OP) over an 11-year period in Grenoble, France. With 1570 filter samples spanning over a decade, this study presents a rather unique dataset investigating how PM10 sources, derived using positive matrix factorisation, and oxidative potential using two assays (ascorbic acid and DTT) evolve over a decade. Relating PM10 sources to oxidative potential highlights the higher OP associated with anthropogenic sources. The authors demonstrate downward trends in PM10 concentrations over this period, as well as a reduction in anthropogenic source contributions to PM10, notably primary traffic and biomass burning. OP measurements also reveal a decrease in volume-normalised AA and DTT derived OP, relating this to the reduction in anthropogenic emissions. Overall, this study is well designed and presented and provides new insights into the longer-term trends of both PM10 sources and OP in Grenoble. This study is well suited for publication in atmospheric chemistry and physics. Below are some specific comments to be addressed.

**Specific Comments:**

Line 54-58 – Whilst I agree that it is important to link sources to oxidative potential, I don't think understanding compositional drivers of OP should be dismissed. Understanding the chemical constituents of specific sources which increase OP is also important, to understand why a particular source has higher OP activity. Especially when considering the intrinsic OP of sources presented in this study, understanding what components within these sources drives OP is also informative for OP abatement.

Line 120 - What was the selection criteria for measuring these specific time points for measurement?

Line 140 – the specific composition of the synthetic lung fluid should be explicitly stated here.

Line 160 – Over what timeframe were filters measured? Were all filters analysed across different time periods, or were all filters across the 10-year period analysed more recently? How were the filters stored over this time? Were any tests performed on filters to observe any compositional variability during storage for over a decade? These are important considerations given the long time over which filters were collected for OP analysis and should be discussed.

Line 171 – Whilst the distributions presented in Figure S2 look reasonable, the R2 for metal imputation is poor in some cases, as presented in Table S1. A time series comparing imputed vs measured metals in the supplement would be useful to better assess the robustness of imputation here.

Line 118 – It is surprising no Fe was measured using ICPMS, given its importance for OP and PMF source factor derivation. Why was no Fe data included?

Line 290-293 – are dust etc mentioned here PMF source factors, or assumed based on composition? Unclear from the discussion here.

Line 415 – it is unclear what "reactivity due to transport" refers to here.

Line 479 – The y-axes here are unlabelled and no description is given in the Figure caption, please add y-axes to these plots and also in Figure 9, and Figures S6, S7, S16, S17.

Line 505 – What is the rate of reduction of wood sales, and did you perform any additional statistics to associate wood burning with wood sales? Seeing a statistical association here would be interesting.

Line 573 – it is unclear why the non-linear models tested here (random forest, multiple layer perceptron) cannot give intrinsic OP values for each source? Please define RF and MLP acronyms here too.

Line 590 – regarding natural sources, no secondary organic aerosols are considered from biogenic sources? So this statement refers only to primary natural emissions? This should be clarified.

Line 598 – OPv is normalised per volume (m-3), not per PM10 mass, as this would be OPm? The word normalised in this context is confusing.

Line 620 – Figures S16 and S17 show interesting trends for OP, and in line with the main aims of the manuscript. I would suggest moving these figures into the main discussion here.

Line 641 – Why are figures not included to show the OPv trends for AA and DTT, similar to S16/S17 for OPm?

Line 643 – As mentioned above, the use of normalised here is a bit confusing.

---

## Author Response (AR1)

We thank the reviewers for their time and valuable comments that helped improve our manuscript's quality. We have answered the reviewers' comments in red and in *blue italic* are the changes included in the manuscript.

**Referee 1**

This manuscript investigated the decadal trends in PM10 sources and oxidative potential (OP) at an urban background supersite in France by 11 years of long-term observations by the method of positive matrix factorization (PMF) analyses, and found the downward trends of OP and the high redox activity of the anthropogenic sources. Such findings are valuable for air quality managements to protect public health. But there are still some defects, and comments are suggested for considering:

Title should be limited as "urban background supersite".

We thank the referee for the proposition. We have updated the title as follows:

*Decadal trends (2013–2023) in PM10 sources and oxidative potential at a European urban supersite (Grenoble, France)*

**Introduction**

Line 64-67: The limitations of short-term studies should be explained more and in depth.

Thank you for your proposition. The introduction aims to highlight the necessity and advantage of long-term study; we only wanted to indicate some applications of the short-term studies and then highlight the advantage of the long-term studies. We feel that discussing in depth the topic proposed by the reviewer would be too wide a discussion and out of focus for our work in this introduction.

**Methods and Materials**

Line 85: What's the representativeness and environmental implications of the urban background supersite?

Thank you for your question. In the EU 2024/2881 Air Quality Directive, a "super site" is designed to collect long-term data to better understand the effects of pollutants on health and the environment, also to measure in the vicinity of sources such as ports and airports, major roads, industries and residential heating. The site in Grenoble has been chosen by the French Ministry of Environment according to these goals, and is satisfying the requirements of the EU Air Quality Directive.
Grenoble is a French Alpine Valley city, located within a basin surrounded by 3 mountain massifs, which represents a typical urban valley environment in the Alpine region. The station "les Frênes" in Grenoble is already for a long time the reference station in the city for the French air quality monitoring network, with this long-term and extensive program for studying the composition of PM in the city. The observation in this site allows to assess the air quality of a typical urban valley environment, where the pollutant accumulation is especially abundant in the winter season. In addition, the residential heating at the site is also known as one of the main emissions in the region.

Line 108-113: Any rainy days for the sampling? The weather is important factor.

Rainy days principally happen in the city in the fall and spring, and the sampling was conducted regardless of weather conditions. However, we did not observe the specific impact of rain events, which is beyond the scope of the trend-focused study.

Lines 129-131: Since the analysis method of HPLC-PAD was changed to another one followed CEN standards from 2017, how to ensure the comparability of the measured data from different methods?

When we changed the method for sugar analysis a few years ago, we performed a large number of comparisons and dual analyses in order to ensure consistency between the results, both for reference material, standards, and real samples.

Lines 133-134: Two models of ICP-MS (ELAN 6100 DRC II and NEXION) were used, so please tell all the quality control and assurance for data comparable.

To ensure our measurements were accurate and consistent, digestion yields and analytical accuracy were verified with reference material ERM-CZ120 (results in agreement with 70-120% recovery). We also tested external quality control solutions to ensure the instrument remained consistent.
In addition, following the validation protocol for our HPLC-PAD method, we performed the dual analysis on the real samples to validate the results between these two ICP-MS models.

Lines 153-154: Details about the external references should be told.

Thanks for your remark. We updated the text as follows (line 155 – 156):

*The batches were standardized using a common control (lab's rooftop filter analysis for every batch) to ensure consistency between batches.*

Line 161: Vertical temperature and humidity were measured only during November 2017 to May 2023, so how to use the data for the 11 years of period?

It is of course out of question to "use the data" for the full period, since there is no real data. This section is intended as an investigation purely concerning the period of time when the data existed, and it is already interesting in itself, providing insightful information for an extended period of time rarely investigated in the literature. Further, even if it is probably the case, we did not propose in the paper the hypothesis that the processes and results obtained for this period most probably apply to the previous period, since it may be debatable and we have to prove that. Finally, in this section 3.3.2, we mentioned that for the analysis of the relationship between PM and temperature vertical gradient, we used only the period where we have both measurements.

Lines 163-165: Can the meteorological station represent the PM10 sampling site?

Thanks for your question. The objective of the vertical temperature measurement is to calculate the temperature gradient. To do so, the measurement should be performed at a site that has different elevation levels accessible, and Bastille Hill is the station nearest the supersite (10km apart).
In addition, Grenoble is a valley city, and as the development of thermal inversion is a large-scale phenomenon encompassing the overall city basin; we can therefore consider that the measurements on Bastille Hill are sufficient to represent the sampling site in terms of the height of the boundary layer.

**Results and Discussion**

Line 297: Why are the concentrations of OM and EC are highest in winter?

Thanks for your question, the concentration of OM and EC is highest in the winter because of 2 principal reasons:
1. Residential heating activities occur during the winter season.
2. The thermal inversion events in the valley city.

Lines 317-318: What is the scientific basis for selecting the three time periods of 2013-2016, 2017-2021, 2022-2023? The reasons for the division must be clearly stated.

Thanks for your remarks. We are sorry for the error in the manuscript; the periods are: 2013-2016, 2017-2020, 2021-2023.
Our purpose is to divide the 11-year dataset into shorter periods to test the stability of the PM sources over time. Our study is one of the first studies that tests this kind of method, and our idea is to separate 11 years into 3 periods to observe the evolution profile of sources. The periods were chosen to create balanced and multi-year segments covering the entire study duration.
We have updated the main text (lines 318-319):

*To evaluate such a phenomenon in our case, we investigated the chemical profile and contribution of PM$_{10}$ sources for three distinct periods (2013-2016, 2017-2020, 2021-2023).*

Lines 356-358: What are the differences in the respective periods of these studies? What are the key differences between the current study and those of Borlaza et al. (2021) and Srivastava et al. (2018)?

The key difference between this study and those of Borlaza et al. (2021) and Srivastava et al. (2018) is principally the study period, study duration, and the input variables (Table below).

|  | Study period | Study duration | Input variables |
|---|---|---|---|
| This study | 2013-2023 | 11 years | Classical |
| Borlaza et al. (2021) | 2017-2018 | 1 year | Organic acids |
| Srivastava et al. (2018) | 2013 | 1 year | HAPs, Alkanes |

The study of Srivastava et al. (2018) observed the different anthropogenic sources (primary and secondary), while the study of Borlaza et al. (2021) principally focused on the SOA sources. These studies are only performed for a 1-year measurement.
On the other hand, our study aims to assess the trend of the main PM10 sources in the city over 11 years, which was conducted with a less extensive set of input data, but with a longer duration than those in the other studies. All in all, we can say that the goals of the studies are quite different.

Lines 395-398: Because the results of this urban site of current study are compared with the rural sites of EMEP (Colette et al., 2021) and Aas et al. (2024), it is suggested to search the literature of similar urban sites for reasonable comparison. If not, please also discuss the differences between rural and urban sites. Moreover, using the rate of change for comparison would be better for explanation.

Thank you for the thoughtful comment. We actually tried to find a long-term trend source study in European background sites, however, for the best of our knowledge, there are virtually no comparable study (with a long-term dataset, including a comparable set of chemical species, encompassing organic markers, etc). We updated in the main text the discussion between the urban and rural sites. We also updated the main text by using the rate of change for comparison (lines 404-412 and lines 424-435):

*The reduction of PM$_{10}$ in Grenoble during this period is significantly larger than that in 30 rural sites of the European Monitoring and Evaluation Programme (EMEP) from 2000 to 2017,*

*which show reductions of $PM_{10}$ from -1.5% to -2.5% (-0.008 to -0.58 µg m$^{-3}$ ) (Colette et al., 2021). However, the results of our study are highly coherent with results from Aas et al. (2024), presenting a reduction of $PM_{10}$ in 2 rural sites in France (La Tardière and Revin) of -3.5% yr$^{-1}$ between 2005 and 2019. The reduction of PM in this Grenoble site, as an urban site, being higher than those at the rural sites, is due to the changes in specific emission activities at the site. While in the rural sites, the PM emission are influenced by long range transport activities, the PM at the urban site is usually largely impacted by different local activities (Borlaza et al., 2022). Further, France is amongst the EU countries with the highest reduction trend, as presented by Aas et al. (2024).*

*The anthropogenic sources, such as primary traffic, sulfate-rich, and biomass burning, display the highest decrease between 2013 and 2023 in Grenoble, with a reduction of 12.9, 6.9, and 5.5% (0.37, 0.25, and 0.13 µg m$^{-3}$ yr$^{-1}$), respectively. The other anthropogenic sources also present significant decreasing trends; however, these trends are much lower (nitrate-rich: -0.11 µg m$^{-3}$ yr$^{-1}$, industrial: -0.02 µg m$^{-3}$ yr$^{-1}$). The downward trends of these anthropogenic sources (mainly traffic, SIA, and industrial) were also underlined for other European urban sites (Colette et al., 2021; Diapouli et al., 2017; Pandolfi et al., 2016) with various approaches. For instance, a similar approach using PMF (albeit without organic markers) was followed by Pandolfi et al. (2016), investigating the Mann-Kendall trend of PMF-derived sources, and reported an almost equivalent downward trend of the sulfate-rich factor of 53% (i.e., 0.53% yr$^{-1}$) between 2004 and 2014 in Spain. The decreasing trends of primary traffic, domestic biomass burning, and industrial emissions are potentially influenced by the reduction in primary emissions due to various abatement strategies (as discussed in the following subsections, notably in 3.3.3 and 3.3.4).*

Lines 429-420: In conjunction with Section 2.2.3, as mentioned above in the Methods, the meteorological analysis only cover 2017-2023, and there is a time mismatch with the PMF source analysis covering 2013-2023, so can the conclusions drawn based on 2017-2023 about the impact of inversion on PM10 and its sources be applicable to the entire 2013-2023 period?

Thanks for your question. Unfortunately, the specific impact of the meteorological inversion is limited to the period of 2017-2023, where the vertical temperature measurement is available. A tentative extrapolation for the unmeasured period is indeed out of reach of our study.

Lines 523-526: Consistent with problems in lines 395-398, please compare with similar urban sites by percentages.

Thanks for your suggestion. We have updated the text (lines 550-555):

*The downward traffic trend observed in this study is consistent with another long-term study (2012-2020) from a rural site in France, which showed a traffic trend of -6.5% yr$^{-1}$ (58% total reduction) (Borlaza et al., 2022). This is aligned with other results of trends for fossil fuel black carbon concentrations in several rural sites in France (Font et al., 2025), or EC concentrations over many rural sites in Europe (Aas et al., 2024). Additionally, our result also agrees with other studies, like that by Pandolfi et al. (2016), which indicated a downward trend of traffic sources in an urban site in Spain, with a reduction of 5.6% yr$^{-1}$ (56% total reduction), which is lower than that of our study.*

Besides local sources, any long-range transport considered?

Thanks for your question. We understand that this question is about traffic sources. Indeed, for our site, we have the assumption that the long-range transport of the traffic source is negligible compared to the local activities.

Figure 2: The air quality guideline could be used for comparison. Some statistics should also be conducted for overall trends.

Thanks for your suggestion. We did mention the air quality guideline in the text (lines 287-289). However, since Figure 3 represents the monthly average, we could not incorporate the air quality guideline, which is established based on the annual and daily mean of PM.

Figure 4: In fact, the seasonal variation is also valuable.

Thanks for your suggestion, the authors agreed that seasonal variations are very important. However, our purpose in this figure is to present the yearly average contribution of sources to PM, assessing the year-to-year change in PM source contributions in the study period. The details of the seasonal and temporal evolution of each source are presented in Figure S4, SI.

Table 1: Is it linear relationship?

Thanks for your question. Yes, as described in lines 278-282, the tendency of sources is evaluated using linear regression (ordinary least squares).

**Referee 2**

This study investigates the trends of PM10 sources and oxidative potential (OP) over an 11- year period in Grenoble, France. With 1570 filter samples spanning over a decade, this study presents a rather unique dataset investigating how PM10 sources, derived using positive matrix factorisation, and oxidative potential using two assays (ascorbic acid and DTT) evolve over a decade. Relating PM10 sources to oxidative potential highlights the higher OP associated with anthropogenic sources. The authors demonstrate downward trends in PM10 concentrations over this period, as well as a reduction in anthropogenic source contributions to PM10, notably primary traffic and biomass burning. OP measurements also reveal a decrease in volume-normalised AA and DTT derived OP, relating this to the reduction in anthropogenic emissions.

Overall, this study is well designed and presented and provides new insights into the longer-term trends of both PM10 sources and OP in Grenoble. This study is well suited for publication in atmospheric chemistry and physics. Below are some specific comments to be addressed.

Specific Comments:

Line 54-58 – Whilst I agree that it is important to link sources to oxidative potential, I don't think understanding compositional drivers of OP should be dismissed. Understanding the chemical constituents of specific sources which increase OP is also important, to understand why a particular source has higher OP activity. Especially when considering the intrinsic OP of sources presented in this study, understanding what components within these sources drives OP is also informative for OP abatement.

Thanks for your suggestion. We agree that the relationship between the composition and OP is also important. However, as highlighted in the studies of our group, trying to relate OP with chemical composition is challenging because of the fraction of unmeasured chemical species with minor components being able to provide a large fraction of the OP. Further, considering the composition is not taking into account the possible synergistic/antagonistic effects among PM components. Finally, we

strongly believe that the OP SA could highlight which emission sectors are more important in health-relevant effects and provide the information to the policy-makers.

We have added a heatmap correlation between PM components and $OP_v$ in the SI to complement our results in OP SA.

Line 120 - What was the selection criteria for measuring these specific time points for measurement?

The daily analysis of trace metal concentrations for specific periods (2013, 2017-2018, 2020-2021) was determined by the funding and objectives of distinct research programs. As mentioned in the acknowledgments, the daily trace element analyses were specifically funded by the QAMECS program. For the other periods, only weekly measurements were available through a long-term monitoring program conducted by Atmo Aura with their own funding.

Line 140 – the specific composition of the synthetic lung fluid should be explicitly stated here.

Thank you for your suggestion. We have updated the text (line 141-143):

*Filter samples are extracted using a simulated lung fluid which is the mixing of Gamble and DPPC (dipalmitoylphosphatidylcholine) solutions, during 1h15 at 37°C, pH 7.4 (Calas et al., 2017), creating a physiological environment for the extraction.*

Line 160 – Over what timeframe were filters measured? Were all filters analysed across different time periods, or were all filters across the 10-year period analysed more recently ?

For organic and inorganic chemical analysis, samples were analysed in the following year after sampling (generally within 3 months). Afterward, the remaining samples were stored at -20°C for long-term conservation.
OP measurements are more recent, and have started on this series since 2016 with the remaining samples. For the period 2016-2024 the OP is measured at the same time as the other chemicals.

How were the filters stored over this time? Were any tests performed on filters to observe any compositional variability during storage for over a decade? These are important considerations given the long time over which filters were collected for OP analysis and should be discussed.

The filters are stored at -20 °C. As mentioned earlier, the 2013–2016 period was analyzed in 2016, and we assume that the filters may have undergone some chemical ageing processes. For the 2016–2024 period, OP measurements were performed simultaneously with chemical analyses. Ageing experiments on OP have shown that a decay of up to 20% in OP levels can occur within the first six months of storage, after which the values reach a plateau. This decay is considerably lower when the filters contain high metal concentrations. Such ageing effects on organic species are inherent to this sampling method and similarly affect the chemical composition measurements. Nevertheless, we are confident that the time series remains robust, given the strong reconstruction of OP levels achieved by the model, even during the bootstrapping phase.

Line 171 – Whilst the distributions presented in Figure S2 look reasonable, the R2 for metal imputation is poor in some cases, as presented in Table S1. A time series comparing imputed vs measured metals in the supplement would be useful to better assess the robustness of imputation here.

Thanks for your comments. We have added the time series in SI.

Line 118 – It is surprising no Fe was measured using ICPMS, given its importance for OP and PMF source factor derivation. Why was no Fe data included?

Thanks for your remark. The exclusion Fe from PMF is because the limit of detection of Fe is high, consequently increasing the uncertainty of Fe concentrations and therefore leading to the S/N of Fe being below 0.2. The species with S/N< 0.2 has to be excluded from PMF, as recommended by the EU Commission, for source apportionment studies.

Line 290-293 – are dust etc mentioned here PMF source factors, or assumed based on composition? Unclear from the discussion here.

Thanks for your comments. We did mention that the dust is estimated using the equation in S2, Eq. (S4) (line 294):
[dust] = 5.6 * ([Ca2+] – [Na+]/26) (Putaud et al., 2004)
At this stage, we only want to present the PM measurement (TEOM-FIDAS) and the chemical mass closure estimation. The PM source apportionment result is presented in section 3.2 below.

Line 415 – it is unclear what "reactivity due to transport" refers to here.

"Reactivity during transport" refers to the chemical and physical aging processes that chemicals undergo in the atmosphere as they travel from their emission source to the receptor site. The upward tendency of the MSA could be related to the increase of the precursor emissions (DMS concentration, for example), or it could be a favorable condition during transportation that increases the photochemistry of MSA.

Line 479 – The y-axes here are unlabelled and no description is given in the Figure caption, please add y-axes to these plots and also in Figure 9, and Figures S6, S7, S16, S17.

Thanks for your comments. We have updated the y-axes.

Line 505 – What is the rate of reduction of wood sales, and did you perform any additional statistics to associate wood burning with wood sales? Seeing a statistical association here would be interesting.

Thanks for your comments. We added the statistics to associate wood burning with wood sales.

Line 573 – it is unclear why the non-linear models tested here (random forest, multiple layer perceptron) cannot give intrinsic OP values for each source? Please define RF and MLP acronyms here too.

Thanks for your comments. We have updated the text (lines 598-599 and lines 603-607):

*In addition, the study revealed good performance of Mutiple Layer Perceptron (MLP) and Random Forest (RF) for the training and testing datasets (Table S10).*

*However, such non-linear models do not provide values for the intrinsic OP, which is basically the regression slope of the regression. Since the objectives of MLP and RF are not to define a "slope" but to better predict OP, therefore, the "slopes" of such models actually constantly vary with the input data to ensure the best performance of the model. Since the OP intrinsic is not defined, these models cannot be selected for the final results at this stage.*

Line 590 – regarding natural sources, no secondary organic aerosols are considered from biogenic sources? So this statement refers only to primary natural emissions? This should be clarified.

Thanks for your remark. Indeed, one secondary organic aerosol fraction is represented with the MSA-rich, characterized by high loading of MSA, a main product of oxidation of DMS. We have updated the text (lines 623-625):

*The natural sources have a negligible intrinsic OP (lower than 0.03 nmol min$^{-1}$ $\mu g^{-1}$ for OP$_{DTT}$ and 0.2 nmol min$^{-1}$ $\mu g^{-1}$ for OP$_{AA}$). These findings highlight the high impact of the anthropogenic sources, verified for the overall period 2013-2023.*

Line 598 – OPv is normalised per volume (m-3), not per PM10 mass, as this would be OPm?

The word normalised in this context is confusing.

Thanks for your remark. We have updated the text (lines 633-634):

*Although $OP_v$ is calculated using $PM_{10}$ concentration, implying that a decrease in $PM_{10}$ concentration generally reduces OP.*

Line 620 – Figures S16 and S17 show interesting trends for OP, and in line with the main aims of the manuscript. I would suggest moving these figures into the main discussion here.

Thanks for your suggestion. We would also like to put these figures in the main text. However, we actually presented 11 figures in the main text, and the paper seems to be dense with two more figures. Therefore, we decided to put them in the SI.

Line 641 – Why are figures not included to show the OPv trends for AA and DTT, similar to S16/S17 for OPm?

Thanks for your suggestion. We have added the $OP_v$ trend to the supplement.

Line 643 – As mentioned above, the use of normalised here is a bit confusing.

Thanks for your comment. We have updated the text (lines 678-679):

*This good agreement partially reflects the influence of the PM mass concentration since these $OP_v$ values are calculated using $PM_{10}$ concentration.*